



# The impact of terrain model source and resolution on snow avalanche modelling

Aubrey Miller[1], Pascal Sirguey[1], Simon Morris[2], Perry Bartelt[3,4], Nicolas Cullen[5], Todd Redpath[5,1], Kevin Thompson[2], and Yves Bühler[3,4]

[1]National School of Surveying, University of Otago, P.O. Box 56, Dunedin, New Zealand
[2]Downer NZ Ltd, Milford Road Alliance, Te Anau, New Zealand
[3]WSL Institute for Snow and Avalanche Research SLF, Flüelastrasse 11, 7260 Davos Dorf, Switzerland
[4]Climate Change, Extremes and Natural Hazards in Alpine Regions Research Center CERC, Flüelastrasse 11, 7260 Davos Dorf, Switzerland
[5]School of Geography, University of Otago, Dunedin, New Zealand

**Correspondence:** Aubrey Miller (aubrey.miller@otago.ac.nz)

**Abstract.** Natural hazard models need accurate digital elevation models (DEMs) to simulate mass movements on three-dimensional terrain. A variety of platforms (terrestrial, drones, aerial, satellite) and sensor technologies (photogrammetry, LiDAR, interferometric synthetic aperture radar) are used to generate DEMs at a range of spatial resolutions with varying accuracy. As the availability of high-resolution DEMs continues to increase and the cost to produce DEMs continues to fall,

hazard modellers must often choose which DEM to use for their modelling. Here we use current state-of-the-art sensor technologies (satellite photogrammetry and terrestrial LiDAR) to generate high-resolution DEMs and test the sensitivity of the Rapid Mass Movements Simulation software (RAMMS) to the DEM source and spatial resolution for simulating a large and complex snow avalanche along Milford Road in Fiordland, Aotearoa New Zealand. Holding the RAMMS parameters constant while adjusting the source and spatial resolution of the DEM reveals how differences in terrain representation between the

satellite photogrammetry and terrestrial LiDAR DEMs (2 m spatial resolution) affect the reliability of the simulation estimates (e.g., maximum core velocity, powder pressure, final debris pattern). At the same time, coarser representations of the terrain (5 m, 15 m spatial resolution) produce simulated avalanches that run too far and produce a powder cloud that is too large, though with lower maximum impact pressures, compared to the actual event. The complex nature of the alpine terrain in the avalanche path (steep, rough, rock faces, tree-less) made it a suitable location to specifically test the model sensitivity to digital surface

models (DSMs) where both the ground and above-ground features on the topography are included in the elevation model. Combined with the nature of the snowpack in the path (warm, deep with a steep elevation gradient) lying on a bedrock surface and plunging over a cliff, RAMMS performed well in the challenging conditions when using the high spatial-resolution 2 m DSM.

## 1 Introduction

Natural hazards like snow, ice and rock avalanches, debris flows and landslides pose risk to people and infrastructure in alpine regions (Badoux et al., 2016; Techel et al., 2016; Dowling and Santi, 2013). While predicting the timing and destructive capacity



of a natural hazard remains an unsolved challenge for researchers, the development of dynamic hazard models to anticipate potential impacts has provided a valuable risk-mitigation tool used in hazard mapping, land planning, for engineering mitigation measures and to reanalyse or back-calculate historic events (Bühler et al., 2022; Baggio et al., 2021; Christen et al., 2010a;

Casteller et al., 2008). Natural hazard modelling of mass transport has evolved rapidly as the physics are better understood and detailed event observations are becoming more readily available for model calibration.

The early development of one-dimensional models (e.g., Bartelt et al. (1999); Salm (1966); Voellmy (1955)) have evolved into more complex two-dimensional and three-dimensional numerical simulations. Some examples of models simulating gravity-driven flows on three-dimensional terrain include RAMMS (Rapid Mass Movements Simulation (Christen et al.,

2010b)), r.avaflow (Mergili, 2020; Mergili et al., 2017), DAN and DAN3D (Dynamic ANalysis (Hungr and McDougall, 2009)), Flow-R (Horton et al., 2013), SamosAT (Snow and Avalanche MOdelling and Simulation – Advanced Technology (Sampl and Zwinger, 2004)), TRENT2D$^{\circledast}$ (Zugliani and Rosatti, 2021) and others (van den Bout et al., 2021; Li et al., 2021; Rauter et al., 2018; Hergarten and Robl, 2015; Medina et al., 2008; Rickenmann et al., 2006). Dynamic models simulate flow characteristics on real-world topography, represented efficiently, but imperfectly, by a digital elevation model (DEM). The DEM is usually

stored as a two-dimensional grid projected in a Cartesian coordinate system, where each cell value is the height above a datum (Claessens et al., 2005; Wise, 2000; Gao, 1997). The technique of topographic data capture and processing, as well as the spatial resolution of the DEM, control the level of detail achieved in elevation models. For instance, DEMs available at a global scale are coarser and miss fine-scale topography. Conversely, very high spatial-resolution DEMs can resolve fine-scale topography but are often limited to smaller spatial extents and impose higher computational requirements for the simulation.

In this context, hazard modellers must judge the most appropriate DEM for the study domain based on availability, cost, currency (the most-recent or before/after an event altered a landscape), season (snow-on vs snow-off), completeness (whether data voids or holes exist the DEM), spatial resolution, accuracy and whether the DEM is a digital terrain model (DTM) or digital surface model (DSM), subsets of the more generic DEM terminology. A DTM is a bare-earth representation of the topography where all above-ground features (e.g., trees, buildings) are removed from the model. A DSM includes both the

ground and above-ground features in the model. The choice of DEM will have implications for the influence of terrain features on flow characteristics, such as runout distance and channel overflowing for debris flows and rock avalanches (Zhao and Kowalski, 2020; Tarolli, 2014; Simoni et al., 2012; Allen et al., 2009), or runout distances, estimated maximum core velocities and impact pressures for snow avalanches (Bühler et al., 2011; Christen et al., 2010b). The effect and propagation of DEM uncertainties in model outputs (Zhao and Kowalski, 2020; Bühler et al., 2011) may be hard to identify and characterise without

access to multiple DEM sources of variable accuracy. While higher-resolution DEMs are expected to represent terrain better, they may not always improve hazard modelling when used at their finest resolution as the model may be overly sensitive to fine-scale features in the case of landslide initiation and snow avalanches (Tarolli, 2014; Christen et al., 2010a). To best balance computational resources on the one hand and resolving appropriately-scaled topography on the other, modellers often resample the DEM to a different spatial resolution from the source resolution. Upsampling a high-resolution DEM to a coarser grid size

for hazard simulation may yield better results, however downsampling to a lower resolution DEM should be avoided (Bühler et al., 2011; Christen et al., 2010a).



## 1.1 Sensitivity of snow avalanche modelling to elevation product

The influence of spatial resolution on snow avalanche simulations has been studied previously (Brožová et al., 2021; Maggioni et al., 2013; Bühler et al., 2011; Christen et al., 2010a), establishing how the simulation is sensitive to the scale of terrain features resolved in the DEM. Morphometric measures of curvature, slope angle, aspect and roughness are all neighborhood functions applied to each cell where decreasing the resolution of the DEM will smooth large-scale terrain features and decrease the roughness of the surface (Brožová et al., 2021; Grohmann et al., 2011; Wu et al., 2008; Sappington et al., 2007; Kienzle, 2003; Gao, 1997; Bolstad and Stowe, 1994). Surface roughness controls friction between the dense core of the mass movement and the surface. Everything else being equal, both Brožová et al. (2021) and Bühler et al. (2011) demonstrated how increased roughness decreases the estimated runout of the dense core flow. Differences in simulations will be more pronounced in paths with confined terrain features such as gullies compared with broad open terrain. The rule-of-thumb has been to use a higher spatial resolution (as fine as 1 m) DEM for rough terrain and wet avalanches, while a coarser (up to 25 m) DEM is sufficient to capture relevant terrain features in smoother more homogeneous terrain, especially for large avalanche events (Bühler et al., 2011; Christen et al., 2010a).

When modelling snow avalanches, a DEM from a summer surface (snow-off) will represent terrain differently from a winter surface (snow-on). Maggioni et al. (2013) ran the same RAMMS simulations on 2 m summer and winter DEMs and found differences in runout length, deposition patterns and core velocities, attributable to both different surface roughness in the paths, as well as the way confined terrain features in the summer surface were filled-in with snow in the winter surface, in turn reducing roughness which resulted in debris spread over a wider area.

Furthermore, while not a problem for coarse global-scale DEMs, when using high spatial-resolution DEMs it becomes important to distinguish between two subsets of the DEM, the digital surface model (DSM) and digital terrain model (DTM). A DSM includes above-ground terrain features such as shrubs, trees and buildings, and is the DEM product typically generated from optical photogrammetry and radar. A DTM distinguishes and removes these above-ground features from the bare-earth representation of the terrain. Based on the DEM capture technology, DTMs are obtained by morphological filtering of the DSM supplemented by direct retrieval of ground elevation below canopies, such as achieved by LiDAR (Light Detection and Ranging). Depending on the point density, DTMs tend to exhibit smooth interpolation where above-ground points are removed. In regions with trees and shrubs a DTM may represent the terrain more smoothly with implications for hazard modelling.

Snow avalanches move over terrain differently depending on the type of avalanche (e.g., slab vs. loose, dry vs. wet), the snow temperature, the snow depth and snow-cover entrainment, among other factors. The use of a DTM creates a more realistic snow-covered surface that may better represent the sliding behavior of a slab avalanche immediately after release. A DSM may introduce unrealistic surface roughness in areas with trees and shrubs and other above-ground features, especially in the release zone, however the DSM may better-represent roughness in the track and runout where above-ground features provide more appropriate friction estimates (Brožová et al., 2021). In rocky alpine regions with few trees or shrubs the DSM and DTM differences will be minor, and the higher roughness in the DSM may better-reflect the terrain over which an avalanche runs.



## 1.2 State of elevation data products

The increased use of airborne and terrestrial LiDAR and optical sensors on RPAS (remotely piloted aircraft systems), airplanes and satellites offers accurate high-resolution (here we consider as 5 m spatial resolution or higher) elevation data at decreasing costs. At the same time projects like OpenTopograhy (Krishnan et al., 2011) improve data-sharing and discovery. Advances in real-time global positioning and data processing workflows for large point cloud and image datasets have also helped generate more, and higher-quality, DEMs. However, most of these datasets exist as a patchwork, both spatially and temporally, with uneven distribution of high-resolution DEMs available globally and derived from a variety of remote sensing platforms and technologies. Many DEMs are generated at a project-level or in response to a natural hazard rather than as part of a regional or national surveying program. They may have good relative accuracy but limited absolute accuracy, thus complicating their use with other datasets. Many alpine regions prone to natural hazards still lack a high-resolution DEM conforming to published accuracy standards. For example, the last national elevation product for Aotearoa New Zealand was generated with aerial photogrammetry in the 1980s as 20 m contours and interpolated into a 15 m DEM (Columbus et al., 2011). Alpine regions may have episodic high-resolution data capture, but the rate of landscape change may render data rapidly obsolete. National programs for delivering standardised high-resolution DEMs exist in a number of countries facing alpine hazards (e.g., Canada, France, New Zealand, Switzerland, United States) but baseline data delivery and renewal timeframes take years and may not prioritize alpine regions (e.g. Toitū Te Whenua / Land Information New Zealand 2021).

Global DEM products (SRTM, ASTER, TanDEM-X, ALOS, see Rodríguez et al. (2006); Courty et al. (2019); Wessel et al. (2018); Takaku et al. (2014) respectively) suffer from the risk of data obsolescence and bring lower overall accuracy and potentially consequential artefacts for hazard modelling (Bühler et al., 2011). The EarthDEM product under development builds off the workflow for generating the ArcticDEM (Porter et al., 2018) and will leverage the archive of high-resolution imagery from the Digital Globe satellite constellation for 2 m DSM generation in regions across the globe. With the number of space-borne optical sensors continuing to increase, the use of high-resolution DSMs processed from stereo satellite images and multi-view imagery is also increasing.

### 1.2.1 Photogrammetry

Satellite photogrammetric mapping (SPM) uses two or more spatially overlapping digital images (classified as stereo, tri-stereo, multi-view) to generate a DSM. Clouds in an image will create data voids or holes in the DSM. Suitable sensor orientation and image contrast improve the DSM and must be considered when generating DSMs from satellite imagery in steep terrain or when the terrain is covered by snow (Eberhard et al., 2021; Shean et al., 2016). Nonetheless, SPM with images from a number of commercial sensors offers a large geographic extent (>400 km$^2$) from a single acquisition capable of delivering a DSM resolution finer than 2 m with sub-metre vertical accuracy (Bhushan et al., 2021; Eberhard et al., 2021; Dehecq et al., 2020; Deschamps-Berger et al., 2020; Shean et al., 2016; Aguilar et al., 2014).

DSMs derived from digital images captured from an airplane and used in aerial photogrammetric mapping (APM) can be more highly resolved (0.5 m) and more accurate (0.15 m) than SPM-derived DSMs but at a higher cost and over a smaller



spatial extent (Bühler et al., 2015; Nolan et al., 2015; Bühler et al., 2012). The development of RPAS photogrammetric mapping (RPM), which includes for structure-from-motion photogrammetry workflows, produces very high-resolution DSMs (0.05 m)
that are as accurate as APM but over smaller spatial extents (1-5 km$^2$ compared with 50-100 km$^2$ for APM) and at a lower cost (Redpath et al., 2018; Bühler et al., 2016). Terrestrial photogrammetric mapping (TPM) has also been used to generate high-resolution DSMs (0.1 m) but with lower accuracy (0.5 m) and spatial extent (0.5-1 km$^2$) as well as greater potential for obstructed terrain where no elevation estimates are generated (Eberhard et al., 2021; Prokop et al., 2015; Thibert et al., 2015).

### 1.2.2 LiDAR

Light detection and ranging (LiDAR) often has the advantage of making more measurements per square meter than photogrammetry, depending on the distance between the sensor and the target. The higher measurement density and possibility of multiple returns through tree canopies allows for DTM generation more commonly than with photogrammetry. LiDAR sensors are most common with aircraft and terrestrial laser scanners, but are also increasingly available for use on a RPAS. Aerial laser scanning (ALS), either from an airplane or helicopter, typically achieves DEM resolutions of 0.5-1 m with a vertical accuracy of 0.1 m
over a extent comparable to APM (50-100 km$^2$) in a single campaign (Reutebuch et al., 2003). RPAS laser scanning (RLS) can generate DEM resolutions as high as 0.1 m and accuracy of 0.01 m over a slightly smaller spatial extent (0.2-0.5 km$^2$) to RPM (Lassiter et al., 2021; Lin et al., 2019). Terrestrial laser scanning (TLS) can achieve very high-resolution DEMs (0.05 m) with accuracy of 0.1 m and a wide range of spatial extents (0.5-5 km$^2$) depending on the number of scanning positions and terrain (Prokop et al., 2015; Sommer et al., 2015; Deems et al., 2013). Although terrain obstruction may complicate capture, TLS can
resolve complex shapes in the terrain, including steep and overhanging features that are not fully visible from above.

While aerial LiDAR is often considered the gold-standard for topographic mapping and hazard modelling, offering both a high-resolution DTM and DSM over large geographic areas, ALS is expensive compared with other platforms (Bühler et al., 2012). Repeat ALS is useful for topographic change detection (Bernard et al., 2021; Booth et al., 2020) but rapid-response acquisitions after major events in remote alpine regions (e.g., Shugar et al. 2021) is often easier with SPM given acquisition
logistics and data processing. An advantage for hazard modellers with access to terrestrial LiDAR is deployment immediately after an event to document landscape change (Bossi et al., 2015; Maggioni et al., 2013; Bartelt et al., 2012; Sovilla et al., 2010).

### 1.3 Goals of this study

Drawing from current state-of-the-art DSMs derived from satellite photogrammetric mapping (SPM) and terrestrial laser scanning (TLS), we look at how differences in terrain representation influence snow avalanche hazard modelling. We simulate a
large avalanche along Milford Road in Aotearoa New Zealand using RAMMS to determine (1) the effects of topographic mapping technologies and (2) the influence of DSM resolution on simulation outputs. The unique combination of terrain (tree-less, rough and steep) and snowpack characteristics (warm, dense, deep) in avalanche paths in Fiordland National Park provide suitable testing conditions to assess the role of terrain representation in the sensitivity of a dynamic hazard model.

After an overview of the study site, we explain the method for generating the DSMs. We then detail the well-documented
avalanche event used in the RAMMS simulations, provide results from the simulations and show how different representations





of terrain altered the simulated avalanche behavior. We then discuss the advantages and disadvantages of the elevation products for this type of simulation, with some lessons for dynamic hazard modelling more broadly.

## 1.4 Study site

The McPherson avalanche path is adjacent to State Highway 94 – Milford Road in Fiordland National Park, Aotearoa New
Zealand (Figure 1). Milford Road connects Te Anau to Piopiotahi / Milford Sound, a popular tourist destination with an estimated 870,000 visitors in 2019 (Milford Opportunities Project, 2021). The highway crosses through alpine terrain and the Homer Tunnel at 927 m a.s.l., which had 300,000 vehicles pass through the Tunnel in 2019 (Waka Kotahi / NZ Transport Agency, 2021). Homer Tunnel is located at the centre of a 17 km stretch of highway with snow avalanche activity primarily affecting the road between June and December.

### 1.4.1 Topography

Pleistocene glaciation in the Fiordland region of southwest Aotearoa New Zealand carved deep valleys linking alpine regions with the Tasman Sea over short distances. Avalanche paths in Fiordland are characterized by large release zones, some with permanent snow, steep tracks with cliffs, and low-angle runout zones in u-shaped valleys. Avalanche paths have average slope angles of 30-35° with cliffs exceeding 75°. The release zones range in size from 8,000 m$^2$ to 860,000 m$^2$ with an average
of 100,000 m$^2$ (Fitzharris and Owens, 1980). Some paths produce plunging avalanches because of the steep tracks (average slope over 50°) where the core detaches from the terrain landing again at the transition to lower-angle runout zones where they quickly lose momentum (Watson et al., 2021; Hendrikx, 2005; Schaerer, 1989; Fitzharris and Owens, 1980).

The geology of Fiordland has important implications for dynamic hazard modelling. The bedrock, predominantly Darran Complex in the study site (Bradshaw, 1990; Wood, 1960), consists primarily of relatively hard gabbro and hornblende-diorite
(Blattner, 1978). The surface topography in avalanche paths is characterised by little soil or vegetation and predominantly exposed bedrock. While avalanches run on bedrock in the tracks, paths often have loose rocks (scree) in the runout zones, which can be entrained in large avalanches. Regular plunging avalanches have also created tarns in some paths (Owens and Fitzharris, 1985) where the change in gradient between track and runout is most pronounced.

The McPherson avalanche path, the focus of this study, has a broad alpine release zone of approximately 60,000 m$^2$ with
a mean slope angle of 39° and primarily south-southeasterly aspect. It is approximately 10 km from Milford Sound and 25 km from the Tasman Sea (Figure 1). The release zone has a mixture of permanent snow and exposed bedrock. The track begins with an over-steepened 150 m cliff, followed by a shelf and another 200 m cliff with slope angles exceeding 75°, and is also comprised of bedrock with some scree. The runout is a valley comprised of rock and grasses with slope angles of 5-15° extending 1 km to Milford Road and the east portal of Homer Tunnel (Figure 2). The only significant vegetation present in
the path (besides alpine grasses) are trees located at the bottom of the path across Milford Road. There is no documentation of the avalanche core reaching the trees, however the powder cloud from a McPherson avalanche in 2004 broke trees at the lowest point in the path. Since comprehensive record-keeping began in 1985, there have been 12 Size 5 McPherson avalanches recorded, half of which were naturally released and half were released with active control.



**Figure 1.** Overview of study site with footprint of satellite imagery and location of ground control points (GCPs) in panel (a), reference maps in panels (b) and (c) and inset map showing the footprint of TLS data extent and McPherson 2020 avalanche fracture line in panel (d). SH94 Milford Road and the alpine weather station are also shown in panels (a) and (d).

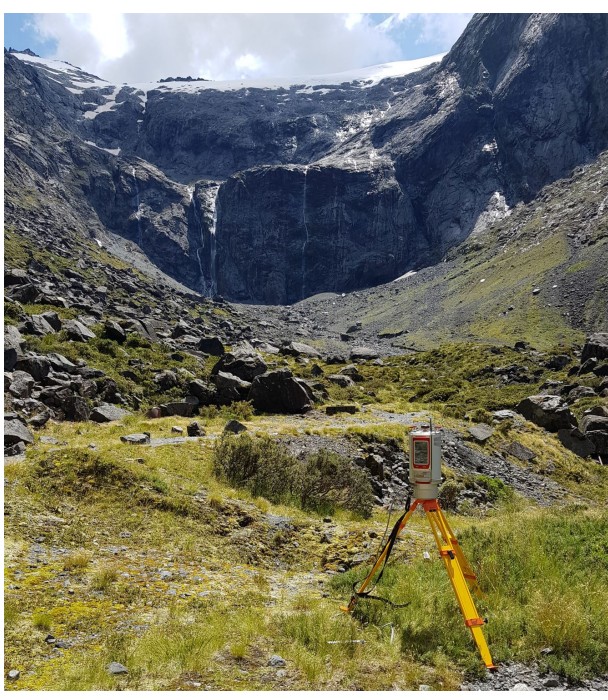

**Figure 2.** McPherson path with release zone visible with snow and the runout in valley below lower cliff face. Riegl VZ-6000 terrestrial laser scanner used in the study shown in the foreground. Homer Tunnel and Milford Road are located behind the camera. Courtesy of Downer Ltd. Milford Road Alliance.

### 1.4.2 Climatic setting

Located on the south-western margin of New Zealand's South Island, and exposed to the prevailing westerly airflow from the Tasman Sea, Fiordland has a maritime climate. Moisture laden airmasses originating in the western half are intercepted by high relief coastal mountain ranges. The highest peak in the road corridor is Mt Christina (2,474 m). This topographic barrier results in substantial orographic enhancement of precipitation. Maximum precipitation rates occur along the coast with a strong eastward precipitation gradient. Mean annual precipitation was 6,716 mm at Milford Sound compared with 757 mm at
Queenstown Airport (75 km to southeast) over the period 1981-2010 (Macara, 2015, 2013). The freezing level is variable and strongly influenced by air mass origin. During winter months, north to north-west flow may bring heavy rain to high elevations. Conversely, south to south-west flow occasionally brings snow to near sea level.

The mean annual precipitation at the East Homer weather station adjacent to Milford Road (874 m a.s.l.) was over 6,209 mm over the period 2011-2020. Over the same period the mean precipitation over the avalanche season (May to December)
was 3,703 mm. The mean annual air temperature was 6.3 °C, over the periods of 1993-2003 and 2010-2020, with a mean avalanche season air temperature of 3.7 °C. The mean air temperature at Cleddau weather station (1,710 m a.s.l, Figure 1) was 1.8 °C, over the period of 2018-2020, with a mean avalanche season air temperature of -0.1 °C. The mean avalanche season





air temperature at Cleddau indicates a mean seasonal freezing level of around 1,700 m a.s.l. Lapse rates in this high relief environment will be influenced by the humid climate.

### 1.4.3 Avalanche mitigation

Avalanche mitigation for paths affecting the highway is handled by a dedicated team of Milford Road Alliance (MRA) avalanche technicians who manage a network of 14 weather stations, over 50 live-feed video cameras and other sensors, including lysimeters, a rain radar, audio and seismic detection, and snowpack sensors. Weather stations and cameras are positioned at a variety of locations from near sea level in Milford Sound to over 2,000 m a.s.l. and including stations located within the typical avalanche release zones between 1600 and 2200 m a.s.l. The MRA technicians also operate a long-range TLS used to measure snow depth in release areas as well as scan avalanche paths from a safe distance after an avalanche event. Active control using explosives is performed with helicopters to mitigate avalanche hazard to road users and infrastructure.

## 2   Methods

This section discusses the data capture and processing workflows for satellite photogrammetric mapping (SPM) and terrestrial laser scanning (TLS) and DSM generation. It then details a case study of snow avalanche modelling used to highlight the sensitivity of the dynamic model to topographic representation.

### 2.1   Digital surface model generation

#### 2.1.1   Satellite stereo imagery workflow

A cloud-free Pléiades-1B image stereo pair was acquired on 10 February 2020 at 10:37 NZST. The panchromatic and multi-spectral (red, green, blue, near-infrared) bands are provided with a spatial resolution of 0.5 and 2 m, respectively, with 12-bit radiometric resolution. The stereo pair had a base-over-height ratio (*B/H*) of 0.38. When processed with Ground Control Points (GCPs), Airbus assesses Pléiades image horizontal accuracy with CE90 (circular error 90%) as high as 0.35 m and vertical accuracy with LE90 (linear error 90%) between 0.8 and 1.2 m depending on slope angle (Airbus, 2021), though higher vertical accuracy has been achieved (Eberhard et al., 2021; Stumpf et al., 2014). The imagery, captured as part of the Pléiades Glacier Observatory (PGO) programme (ISIS/CNES), covered an area of 459 km$^2$ of predominantly alpine terrain in Fiordland National Park. Acquisition was timed for late summer to minimise seasonal snow in the images.

The Pléiades image processing followed the workflow detailed in Eberhard et al. (2021) and included triangulation in ER-DAS Imagine v2018 and surface restitution in NASA Ames Stereo Pipeline v2.7 (ASP, (Beyer et al., 2018; Shean et al., 2016)). We collected 10 GCPs over the imaged area alongside 29 tie points to triangulate the stereo-pair with refined RPC modelling (Figure 1). The GCPs were collected in 30-mn fast-static mode for post-processing against permanent GNSS stations. Fixed solutions achieved centimetre-level precision, were converted to NZGD2000 and projected to NZ Transverse Mercator 2000 (EPSG:2193) using the rigorous deformation model provided by Toitū Te Whenua/Land Information New Zealand (LINZ).





We assessed the quality of the triangulation using leave-one-out-cross-validation (LOOCV) (Eberhard et al., 2021; Sirguey and Cullen, 2014), which yielded a 0.45 m CE90 and 0.63 m LE90.

Dense stereo-matching on the panchromatic bands at 0.5m resolution was performed with a hybrid global-matching approach in ASP (Eberhard et al., 2021; Beyer et al., 2018; d'Angelo, 2016; Hirschmuller, 2008). DSMs were generated from the full-resolution point cloud at 2 m, 5 m and 15 m to the extent of the McPherson study site using the *point2dem* tool in ASP (Beyer et al., 2019) for use in the avalanche simulations. A point cloud was converted from the 2 m DSM for use in co-registration of the TLS data (see next section). We also generated a map of ray intersection errors from stereo-matching in ASP, which is a

measure of the minimal distance between rays for pairwise stereo and can be used to indicate the quality of the match between images, including the identification of areas of poor contrast where surface restitution is compromised (Eberhard et al., 2021). All elevation products use the NZTM2000 coordinate system with height above ellipsoid (HAE). For brevity, we will hereafter refer to the Pléiades-derived DSM products as SPM (satellite photogrammetric mapping).

### 2.1.2  Terrestrial laser scanning workflow

A Riegl VZ-6000 terrestrial laser scanner was used to scan summer topography around Homer Tunnel on the Milford Road. The ultra-long range scanner has a manufacturer-assessed accuracy of 0.015 m with an operational range exceeding 6 km. A composite point cloud of the Homer Tunnel area was created from 15 individual scans captured between 6 October 2019 and 27 February 2021 using Riegl's RiSCAN Pro v2.14 software. Multiple scans were needed to cover the entirety of the avalanche path and surrounding terrain as well as account for the steep topography and occluded terrain from the valley floor.

Scans of the alpine terrain were acquired between February and March to minimize seasonal snow. Areas of permanent snow in the upper release zone captured in more than one individual scan were manually cleaned to remove the points associated with the higher surface, so the composite scan represented the minimum permanent snow level between 2019 and 2021. Scans of the terrain adjacent to the road occurred at other times of year but only when free of snow. A total of 323 check points on objects visible in multiple scans with an average of 22 check points per individual scan were used to merge the individual scans

into the composite point cloud. The mean deviation between checkpoints from scanning positions was 0.057 m. This point cloud (3.3 billion points) had varying point densities within the composite cloud so was thinned to an average point spacing of 0.15 m, or total of 338,280,500 points.

While the TLS point cloud was accurate in relative terms and could be used in hazard modelling without coordinate transformation, absolute georeferencing was required to compare simulations and was achieved by co-registration to the SPM point

cloud. To minimise the influence of unreliable points in the SPM point cloud, the ray-intersection error map was used to exclude points in areas of large intersection error generally corresponding to low image contrast (Sirguey, 2019). Specifically, a 12-cell rectangular median low pass filter was applied to the intersection error map. Points located in cells with an intersection error less than 2 m were sampled to retrieve a mean intersection error (0.13 m), which was used as a threshold for identifying pixels of low-contrast. The segmented SPM points were visually checked for fit with shadows and other areas of low contrast

(e.g., bright snow). While only segmented points were used for the co-registration of the TLS point cloud, the full DSMs used in the simulations included all points, to reflect the overall terrain representation in steep terrain with areas of variable



**Table 1.** Description of the Digital surface models (DSMs) used in RAMMS simulations and their derivations.

| DSM product name | Platform | Sensor | Date of acquisition | Sensor technology | Source DSM spatial resolution | DSM spatial resolutions used in simulations |
|---|---|---|---|---|---|---|
| SPM | Satellite | Pléiades-1B | February 2020 | Photogrammetry with single stereo pair | 2 m | 2 m, 5 m, 15 m |
| TLS | Terrestrial | Riegl VZ-6000 | October 2019 to February 2021 | Long-range laser scanning; composite from 15 scans | 0.5 m | 1 m, 2 m, 5 m, 15 m |
| NZSoSDEM | Aircraft | Zeiss RMK A 15/23 | December 1988 | Manual stereo-plotting; interpolation from 1:50,000 scale 20 m contours by Columbus et al. (2011) | 15 m | 5 m, 15 m |

contrast. Initial alignment of the TLS and segmented SPM point clouds was done manually in CloudCompare v2.10.2, and refined co-registration was achieved with the Iterative Closest Point (ICP) algorithm (Rusinkiewicz and Levoy, 2001). The co-registered TLS point cloud was interpolated into a full-resolution 0.5 m DSM as well as 1 m, 2 m, 5 m and 15 m DSM using

the *point2dem* tool in ASP. The area of the 2m DSM was 5.31 km$^2$.

### 2.1.3   National elevation product workflow

In addition to state-of-the-art DEMs we also incorporated the current national elevation product into the analysis. The last nation-wide topographic mapping campaign produced a 1:50,000 scale topographic map with 20 m contours generated from analogue aerial photogrammetry in the 1980s. Images of the study site were captured in December 1988 and contours gen-

erated with stereo-plotting. Contours were then interpolated into a 15 m DEM by Columbus et al. (2011) and released as the NZSoSDEM v1.0 product, which had a root mean square error (RMSE) of 7.1 m and mean absolute error (MAE) of 5.1 m compared with the national geodetic control network. For the purposes of the analysis, in addition to the full-resolution 15 m DEM, the NZSoSDEM was also downsampled to 5 m using cubic convolution for use in avalanche simulations. The DSM products used in the RAMMS simulations are summarized in Table 1.

### 2.1.4   Hole-filling

High-resolution DEMs often contain holes, or data voids, which can cause issues with hazard modelling. Holes in photogrammetry-derived DSMs are common where no stereo match is possible because of poor contrast (in case of SPM DSMs). Occluded terrain lacking line-of-sight to the sensor also translated to holes in both SPM- and TLS-derived DSMs. Smaller holes were filled with interpolation based on neighbouring cell elevation values, but larger holes were not filled to avoid misrepresenting

this terrain. Holes were filled consistently by applying a max hole-filling threshold of 100 m$^2$, operationalized by adjusting the number of cells to fill in the ASP *point2dem* tool based on the resolution of the DSM (e.g. the 5 m DSM had a 20 cell


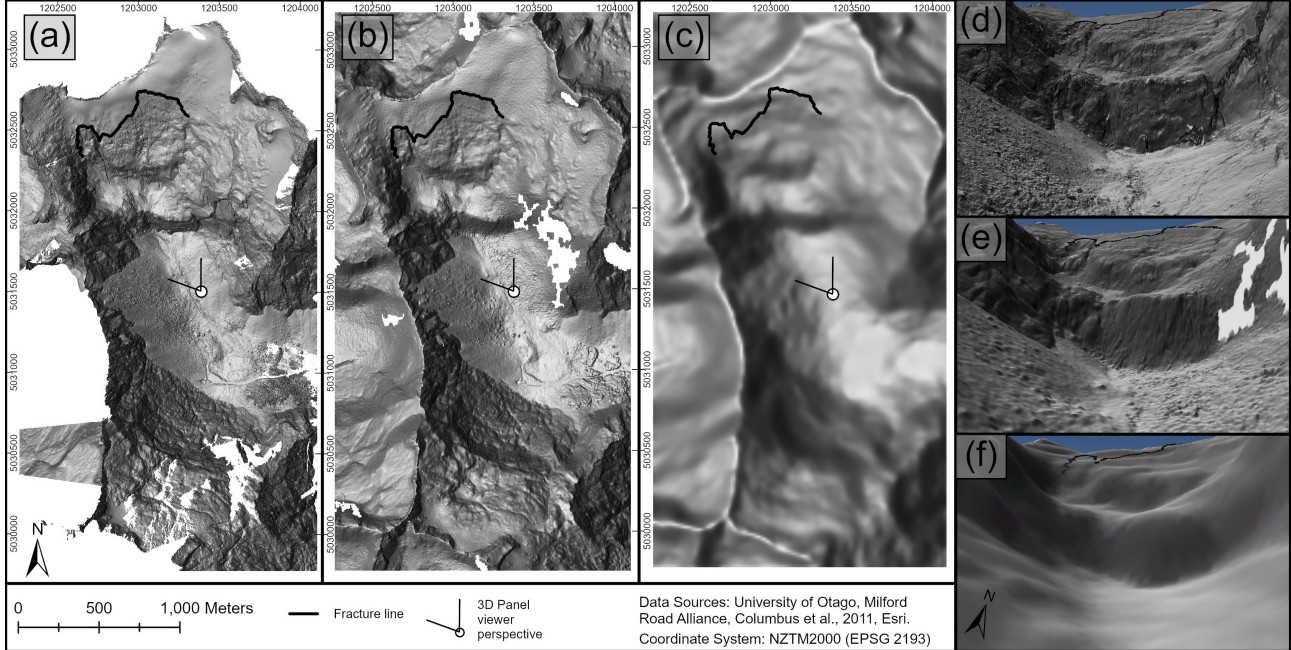

**Figure 3.** Full-resolution DSMs for TLS surface at 0.5 m in panel (a), SPM surface at 2 m in panel (b) and NZSoSDEM surface at 15 m in panel (c). The 3D views focus on lower cliff face and runout zone of McPherson path, corresponding with TLS 0.5 m surface in panel (d), SPM 2 m surface in panel (e) and NZSoSDEM 15 m surface in panel (f). The fracture line from avalanche event is denoted with a black line.

fill limit). For all generated surfaces, the search radius was set to 1.2 to create a smoother transition to hole-filled elevation estimates by using some points on the edge of neighbouring cells. The search radius is a function of the output cell size, where $search\,radius = 1.2 \times DSM\,resolution$.

To assess the maximum area filled with interpolated cells we produced a full-resolution 0.5 m DSM from the TLS point cloud with no holes filled and compared this with the 0.5 m DSM with holes up to 100 m$^2$ filled. Within the study site domain, 3.32% of the area was filled by holes for the 0.5 m DSM and coarser resolution DSMs required less hole-filling. Figure 3 shows the full-resolution DSM for each surface (TLS, SPM, NZSoSDEM). Holes were present in the TLS and SPM surfaces, while no holes were present in the NZSoSDEM surface.

## 2.2 Avalanche event and simulation calibration

The McPherson avalanche was released with explosives dropped from a helicopter on 19 September 2020. The event was documented with videos from the ground and on-board the helicopter, satellite imagery, an infrasound array located down valley from the path Watson et al. (2021), and three TLS scans of the release area and valley debris from the highway the following morning. The entire crown wall and most of the upper path was scanned (mean density of 19 points/m$^2$) with four smaller areas selected for increased point density (mean density of 468 points/m$^2$) to resolve the snowpack structure at the


crown wall. Due to safety of the operator, only an oblique view of the debris was possible. An enhanced 3 m resolution 4-Band (Red, Green, Blue, Near-infrared) PlanetScope satellite image (Planet Team, 2020) acquired the following morning was used to manually digitise the debris in the valley. The terminus of the debris reached a large rock visible in both the imagery and the TLS.

The fracture depth (slope-perpendicular) was calculated from the TLS scan. The fracture ranged from 1,550 to 1,780 m a.s.l. with a total length of 1,346 m. A custom python script was used to calculate crown wall height profiles along the fracture, which had a mean fracture depth of 1.53 m (range 0.003 m to 2.95 m). The first visible slab failure occurred over 600 m from the explosive charge at the bottom of the first of three distinct release areas. They released three, five and six seconds after the detonation. The release areas were manually digitised based on the fracture line and video evidence. Despite the large release
area (total of 117,093 m$^2$), this only accounted for approximately 20% of the available terrain in the release zone.

An alpine weather station at 1,700 m a.s.l. and located 600 m from the release area had a snow depth of 3 m and mean snow density of 400 kg/m$^3$ at the time of the avalanche event. The freezing level dropped to 800 m a.s.l. for a period leading up to the most recent storm but then rose again to 1,600 m a.s.l. where it stayed until the event. This meant there was a steep gradient of available snow for entrainment in the model from >3 m in the upper release area to no snow in the runout in the valley, which
is common in Fiordland avalanche paths. The mean snow temperature in the top 1.5 m of snowpack at the weather station was -0.8 °C at the time of the avalanche, with a mean temperature of -1.3 °C in the remaining snowpack. For more detail on the weather and snow conditions in the month leading up to the event, see Watson et al. (2021).

The avalanche was simulated using the scientific (extended) version of RAMMS (Bartelt et al., 2016; Valero et al., 2016) and calibrated using the 2 m TLS DSM. The simulation used a two-layer mixed model simulating both the dense core (Buser
and Bartelt, 2015) and the powder cloud (Bartelt et al., 2016). RAMMS implements a DEM as a function $Z(X,Y)$ in a Cartesian coordinate system where the independent variables $x$ and $y$ give the arc length along the surface, and where the $z$-coordinate is perpendicular to the slope (Christen et al., 2010b). RAMMS resamples the input DEM if the simulation grid size is smaller or larger than the input DEM resolution. Instead of resampling the DEM in RAMMS by altering the grid resolution of the simulation to be either finer or coarser than the input DEM resolution (Maggioni et al., 2013; Christen et al., 2010a), we ran
each simulation with the associated grid size to the input DSM resolution. In other words, the TLS and SPM surfaces used in the RAMMS simulations were not resampled after the initial interpolation from the point cloud. This means that a greater number of elevation measurements are used to interpolate coarser DSMs and better-represent the true terrain compared with resampling after the initial interpolation to a coarser resolution. We did however resample the NZSoSDEM into a 5 m DSM from the full-resolution 15 m DSM for the purposes of an additional comparison resolution using cubic convolution.

RAMMS allows for more than one snow layer to correspond to different densities and temperatures with their own independent depths. We used two snow layers to better represent the structure of the snowpack based on the temperature profile of the snowpack at the nearby weather station and from the TLS scan of the remaining snow in the release after the event (Figure 10). RAMMS adjusts snow depth in the path based on the slope angle and curvature and with an elevation gradient to more realistically represent snow distribution on alpine terrain, as less snow tends to accumulate in very steep terrain (Sommer et al.,
2015). The maximum snow depth was set to 3 m (two layers each with depth of 1.5 m) at 1700 m a.s.l. tapering to a depth of





0 m at 950 m a.s.l. For review of the extended RAMMS modules, see Buser and Bartelt (2015, 2009) on particle movement, Fischer et al. (2012) on curvature effects, Bartelt et al. (2015) on cohesion within the core, Valero et al. (2016, 2015) on warm, wet avalanches, and Ivanova et al. (2022); Bartelt et al. (2016); Dreier et al. (2016) on the powder cloud. While the snow temperature and density values recorded in the snowpack at the time of the avalanche are usually associated with wet avalanches, 340 the terrain in the McPherson path produced a powder cloud nonetheless.

Based on video evidence of the avalanche and satellite imagery showing the debris pattern in the valley, as well as the TLS scans, the simulation matched the documentation for the powder cloud behavior (frontal velocity and areal extent) and final debris pattern. The simulation was then re-run using the TLS 0.5 m, 1 m, 5 m, and 15 m DSMs, the SPM 2 m, 5 m, and 15 m DSMs and the NZSoSDEM 5 m and 15m DSMs with all snow parameters remaining constant. In other words, the only 345 difference between simulations was the elevation model source and/or resolution. The 0.5 m TLS is not included in the results presented below because the processing requirements for the size of the avalanche made it impracticable. In addition to testing the different DSM sources and resolutions we also ran simulations with a deeper snowpack to mimic a smoother winter surface where small terrain features would be covered by snow. The lower depth value was increased from 1.5 m to 3 m (total depth of 4.5 m) at 1,700 m a.s.l. The lower layer temperature remained at -1.3 °C. All other parameters were held constant and both the 350 TLS and SPM 2 m simulations were re-run with the additional snow.

### 2.3 Differences in topographic representation

To compare the differences in topographic representation between the surface models, DEMs of difference (DoDs) were calculated between the DSM sources. Typically used for detection of landscape change (e.g., Berthier et al. 2014; Lague et al. 2013; Wheaton et al. 2009), here the DoD was used to identify areas of the avalanche path with different elevation values 355 between DSMs from a range of possible drivers. These include acquisition timing (e.g., presence of seasonal snow), angle to the sensor and spatial resolution, artefacts in the interpolation of holes in the DSM, or uncertainty in the stereo matching for the photogrammetry-derived DSM, which all could affect how topographic features important for hazard modelling, like gullies or cliff faces, are represented in the surface model. The DoDs, expressed as $\Delta DEM = Z_1 - Z_2$ where $Z_1$ and $Z_2$, correspond to the DSM with the same resolution but different source, for example $TLS_{2m} - SPM_{2m}$. The results of the DoDs focus on 360 terrain differences between the TLS and SPM DSMs which were co-registered and where fine-scale terrain feature differences will be most pronounced compared with the NZSoSDEM, owing to their higher spatial resolutions.

### 3 Results

The McPherson avalanche released in three distinct areas in quick succession. Snow from the first and smallest release area was first to flow over the lower cliff band, however the flow from the second and third release areas generated the greatest 365 core and powder cloud velocities. It went over the upper cliff with average slope angle of 64°, crossed a shelf with an average slope angle of 13° and then plunged over the lower cliff, with an average slope angle of 76°, ejecting the core into the air (see Figure 4). Despite the warm snow temperature, a large powder cloud was generated. The avalanche reached the valley




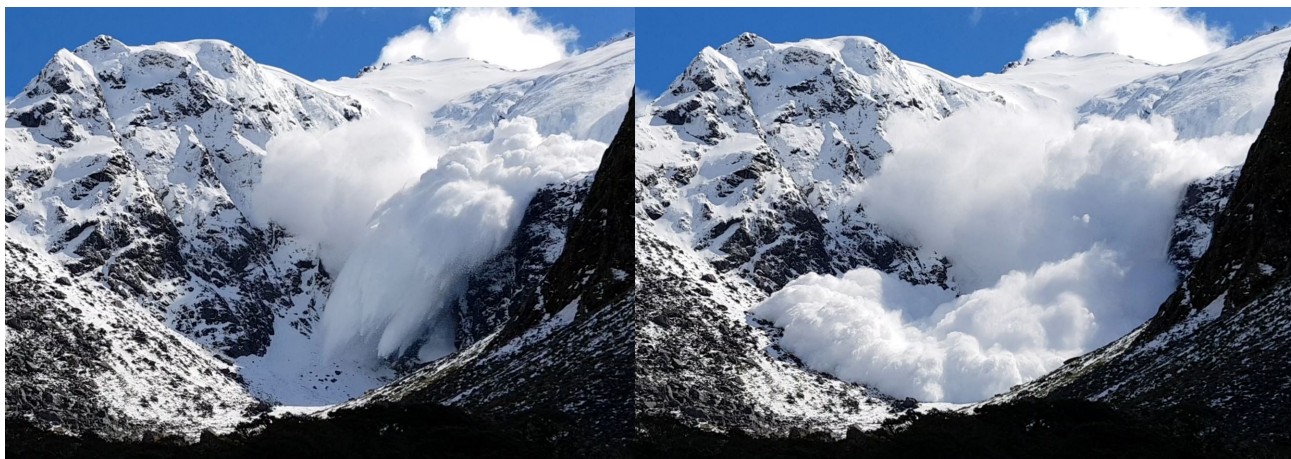

**Figure 4.** Images of McPherson avalanche event as the core runs over the lower cliff (left) from initiation) and powder cloud accelerates into the valley (right) taken from video documentation of event. Courtesy of Downer Ltd. Milford Road Alliance.

floor approximately 39 seconds after the explosive charge detonated. The core travelled over 500 m down the valley before terminating at a large rock visible in both the TLS point cloud and satellite imagery. The average slope angle between the lower
cliff and the terminus rock was 13°. The powder cloud reached maximum velocity in the upper portion of the valley but was visible as it lightly drifted past the Homer Tunnel entrance, 500 m further than the core.

### 3.1 Calibration simulation

The TLS 2 m calibration simulation matched the core flow and powder cloud behavior provided by the documentation. Simulation results are summarized in Table 2. The maximum estimated deposition depth was 7.6 m and the maximum estimated
erosion depth in the track was -2.3 m. The total volume calculated inside the deposition area identified by the TLS and PlanetScope image was 94,934 m$^3$, accounting for 99% of the total simulated deposition below the lower cliff. The simulated avalanche had a release volume of 179,254 m$^3$ (similar to the 1968 Swiss *In den Arelen* avalanche Christen et al. (2010a)) and release mass of 71,702 t. The total core mass was estimated at 119,670 t. By the Canadian avalanche size classification used internationally (Moner et al., 2013; McClung and Shaerer, 1980), the simulation suggested the avalanche was a Size 5. The
avalanche core was estimated to have a maximum flow height of 51 m and maximum pressure of 1,383 kPa. The powder cloud was estimated to have a maximum height of 330 m and pressure of 96 kPa. Powder pressures exceeding 5 kPa covered over 21.8 ha of the path and adjacent valley walls and were greatest at the bottom of the lower cliff.

The maximum core velocity, which is calculated as the mean of the maximum velocity profile values for a given cell, was 78 m s$^{-1}$ or 280 km h$^{-1}$. Figure 5 shows the estimated maximum core velocity and maximum powder pressure. Core velocities
were greatest as the core flowed over the top of the lower cliff, while the powder cloud velocities were greatest as the core splashed over terrain at the bottom of the cliffs rapidly pushing air in front of the core and accelerating away from the cliff. The simulated avalanche hit the valley bottom 41 seconds after the explosive charge (two seconds slower than video evidence),

**Figure 5.** Results of TLS 2 m calibration simulation showing maximum estimated core velocity (top) and powder cloud pressure (bottom). The core velocity saturates red at 40 m s$^{-1}$ while the powder pressure saturates red at 5 kPa. The track profile is shown in the plots for context of the location of the cliff, shelf and runout.

so these velocity estimates are conservative. The plunging nature of the avalanche over the lower cliff adds complexity to the modelling. The sharp transition in the upper path between the steep terrain into the lower angle shelf accelerated the core such

that it went airborne over the second, steeper cliff. The complex relationship between slope angle and the core velocity can be seen with the plot in Figure 5 that shows the influence of the slope transitions in the upper and lower path. When the core splashed into the valley, the greatest powder pressures were estimated.





**Table 2.** Results of RAMMS simulations for DSM resolution by elevation source: terrestrial laser scanning (TLS), satellite photogrammetric mapping (SPM), and analogue aerial photogrammetry (NZSoSDEM) and the coefficient of variation for all simulations. All simulation parameters held constant after calibration based on TLS 2 m DSM (bolded). Release area (RA) has been abbreviated.

| | TLS surface simulation | | | | SPM surface simulation | | | NZSoSDEM surface Simulation | | Coef. of variation |
| | 2 m* | 1 m | 5 m | 15 m | 2 m | 5 m | 15 m | 5 m | 15 m | |
|---|---|---|---|---|---|---|---|---|---|---|
| Core max. height (m) | **51** | 28 | 32 | 19 | 28 | 19 | 18 | 19 | 11 | 46% |
| Core max. velocity (m s$^{-1}$) | **78** | 72 | 68 | 67 | 59 | 66 | 66 | 63 | 64 | 8% |
| Core max. pressure (kPa) | **2,767** | 2,330 | 2,057 | 2,020 | 1,565 | 1,970 | 1,955 | 1,806 | 1,826 | 17% |
| Powder max. pressure (kPa) | **96** | 43 | 60 | 26 | 70 | 54 | 26 | 39 | 27 | 48% |
| RA1 $\mu$ slope angle (deg) | **42** | 42 | 41 | 39 | 40 | 40 | 39 | 30 | 31 | 12% |
| RA2 $\mu$ slope angle (deg) | **34** | 35 | 34 | 34 | 34 | 34 | 34 | 31 | 31 | 5% |
| RA3 $\mu$ slope angle (deg) | **37** | 37 | 37 | 36 | 37 | 37 | 37 | 35 | 36 | 2% |
| Total RA area (m$^2$) | **117,093** | 119,017 | 115,862 | 113,513 | 116,241 | 115,751 | 113,586 | 111,979 | 110,501 | 2% |
| Total RA snow vol. (m$^3$) | **179,254** | 182,327 | 176,924 | 174,470 | 177,553 | 177,128 | 177,506 | 171,103 | 169,800 | 2% |
| Total core vol. (m$^3$) | **267,610** | 256,020 | 280,461 | 310,592 | 269,889 | 265,866 | 299,830 | 284,320 | 331,345 | 9% |
| Total eroded vol. (m$^3$) | **309,643** | 261,375 | 352,021 | 409,092 | 329,729 | 315,353 | 402,338 | 315,929 | 414,151 | 15% |
| Release mass (t) | **71,702** | 72,931 | 70,770 | 69,788 | 71,021 | 70,851 | 71,002 | 76,997 | 76,410 | 4% |
| Core mass (t) | **119,670** | 114,778 | 125,677 | 139,474 | 120,794 | 119,145 | 134,426 | 126,515 | 148,659 | 9% |
| Powder mass (t) | **6,350** | 4,568 | 7,165 | 5,225 | 7,017 | 6,308 | 5,019 | 5,699 | 4,547 | 17% |
| Max. deposition (m) | **7.64** | 12.87 | 5 | 3 | 6.9 | 3.35 | 2.73 | 4.81 | 4.2 | 57% |
| Max. erosion (m) | **-2.34** | -2.46 | -1.81 | -1.78 | -2.06 | -1.75 | -1.86 | -1.87 | -1.8 | 13% |
| Distance from terminal rock (m) | **28** | -245 | 316 | 474 | 205 | 469 | 557 | 547 | 482 | 87% |
| Deposition vol. in debris area (m$^3$) | **94,934** | 55,067 | 57,933 | 37,058 | 72,584 | 44,015 | 31,208 | 27,395 | 42,503 | 42% |
| Total deposition vol. in runout (m$^3$) | **95,872** | 54,202 | 128,190 | 159,390 | 117,080 | 139,513 | 155,700 | 125,875 | 140,175 | 17% |
| Proportion of deposition vol. in debris area to TLS 2 m reference | - | 58% | 61% | 39% | 76% | 46% | 32% | 29% | 45% | 34% |
| Proportion of deposition vol. located in debris area | **99%** | 102% | 45% | 23% | 62% | 32% | 20% | 30% | 30% | 54% |

* reference simulation calibrated against avalanche event documentation





## 3.2 Simulation differences by DSM resolution

This section outlines how using the same DSM source (TLS-derived) but changing the resolution (1 m, 5 m, 15 m) affected
the simulated avalanche behavior. The surface difference (estimate of erosion and deposition) for each simulation is shown in
Figure 6 grouped by DSM resolution. The general pattern is that coarser-resolution simulations (5 m and 15 m) resulted in the
avalanche core traveling too far down the valley and thinning out in the deposition area, the result of the DSM interpolation
smoothing small changes in topographic relief. With larger cell sizes (coarser DSMs) the release areas are smoother resulting
in lower initial snow volumes. However, increasing the cell size increased the volume of snow entrained in the core and the
total eroded snow volume. Likewise, the release mass was smaller with coarser DSMs but the total core mass was larger (2
m was 119,670 t; 5 m was 125,677 t; 15 m was 139,474 t). The mass of the core that is converted to the powder cloud had
no linear relationship with DSM resolution. However the maximum estimated powder cloud pressure (kPa) did decrease with
increasingly coarse DSM resolution. The exception was the 1 m DSM which generated a smaller powder cloud since much
less mass was ejected over the lower cliff.

The highest-resolution 1 m TLS simulation showed the avalanche stopping short of the real avalanche (245 m before the
terminus rock), with much of the initial entrained snow getting stuck on the shelf above the lower cliff. The 1 m simulation
resulted in the deepest estimated deposition (12.87 m) as snow piled up on the shelf. While the snow volume in the release
areas was greater than any other simulation, owing to the higher surface roughness, less snow was eroded and the core mass
was lower than with the 2 m, 5 m, and 15 m simulations. The powder cloud was also smaller with less mass converting from
the core to powder cloud and dissipating midway down the valley.

The 5 m TLS simulation resulted in the core traveling too far down the valley with the toe of debris 316 m beyond the
terminus rock. The maximum estimated debris depth was 5 m and the maximum estimated erosion was -1.81 m. While the
volume of snow in the release areas was lower than in the 2 m simulation, the 5 m simulation had a larger core, eroding more
snow. The total volume of snow in the documented debris area (57,933 m$^3$) was 39% less than in the reference 2 m simulation
as the core maintained momentum further down the valley. Only 45% of the total debris volume was located in the documented
area. The powder cloud covered a larger area compared with the 2 m reference simulation with powder reaching Homer Tunnel
at 20 m s$^{-1}$ and extending well past the highway. At the same time, the powder cloud had a lower estimated maximum pressure
compared with the 2 m reference simulation.

The 15 m TLS simulation also resulted in the core traveling too far down the valley with the toe of the debris 474 m from
the terminus rock. The maximum debris height was only 3 m and erosion -1.78 m. The 15 m simulation had the largest total
eroded volume (409,092 m$^3$) and core mass (139,474 t). The total debris volume in the documented area was 37,058 m$^3$, 61%
less than the 2m reference simulation, resulting in only 23% of the total debris volume located inside the documented area as
most debris was spread further down valley. The powder cloud covered a larger area than the 2 m or 5 m simulations reaching
Homer Tunnel with velocities of 30 m s$^{-1}$ and extending well past the highway. The powder cloud had the lowest maximum
pressure compared with the other TLS simulations. It also maintained a more uniform shape around the core in the valley, with
little lateral spreading to edges seen in the 1 m, 2 m, and 5 m simulations. Figure 7 compares the deposition pattern differences


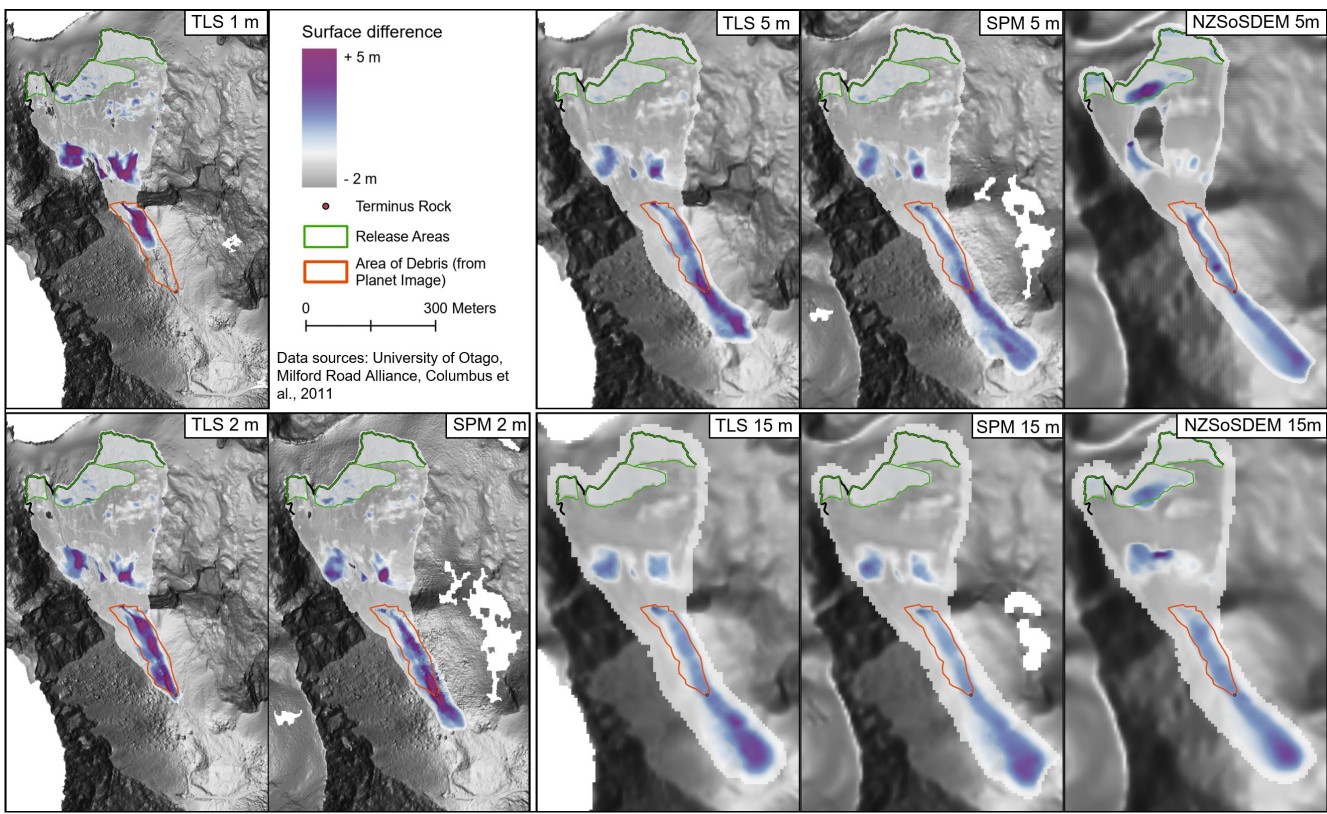

**Figure 6.** Composite of simulation outputs for TLS, SPM and NZSoSDEM DSMs showing surface difference–areas of estimated erosion and deposition. Outputs are grouped by resolution. TLS 1 m is in panel (a), TLS and SPM 2 m are in panel (b), TLS, SPM and NZSoSDEM 5 m are in panel (c) and TLS, SPM and NZSoSDEM 15 m are in panel (d). The fracture line, release areas and debris area are shown in each map. The event was calibrated on the TLS 2 m DSM in panel (b).

between TLS 1 m, 2 m, 5 m, and 15 m simulations as a profile from the lower cliff face to the furthest reach of simulated debris.

### 3.3 Simulation differences by DSM source

This section compares simulation results between DSM source (TLS, SPM and NZSoSDEM). The 2 m TLS and 2 m SPM simulations were similar in a number of ways (see Figure 6, panel (b) and Table 2). Overall, the 2 m SPM DSM best-matched the pattern of surface difference of the 2 m TLS calibration simulation, however there were clear differences. The debris in the 2 m SPM simulation overshot the terminus rock by 205 m and left less debris on shelf between cliffs. The 2m SPM simulation had a maximum deposition depth of 6.9 m and erosion of -2.06 m. Figure 7 compares the surface difference profiles between 435    the 2 m TLS and SPM simulations and shows the SPM core traveling further down valley but with a similar deposition pattern.



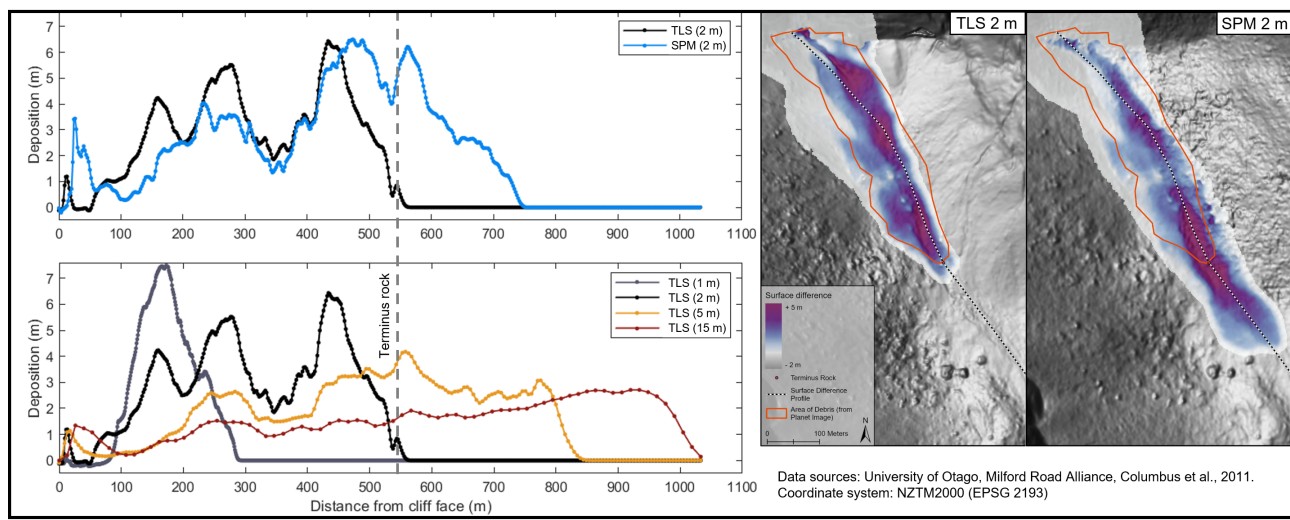

**Figure 7.** Profile of simulated debris from lower cliff face through runout zone between TLS and SPM 2 m simulations (top left) and between TLS 1 m, 2 m, 5 m, and 15 m simulations (lower left). Examples of simulated debris patterns for TLS and SPM 2 m simulations (right side) show how well debris pattern matches documentation (orange polygon).

While the release volumes and total core volumes were similar between the 2 m simulations, the 2 m SPM simulation eroded a greater volume of snow (329,729 $m^3$ compared with 309,643 $m^3$) and converted more mass to powder (7,017 t compared with 6,350 t). The 2 m SPM simulation deposited 24% less volume inside the documented release area compared with the 2m TLS calibration simulation. Only 62% of the total deposition volume in the valley was located inside the documented debris area. The core and powder velocities were slower and the maximum core height and maximum powder pressures were lower than the 2 m TLS simulation.

There were 5 m simulations run on the TLS, SPM and NZSoSDEM surfaces, which exhibited substantial differences (Figure 6, panel (c)). The NZSoSDEM simulation had the longest run beyond the terminus rock (547 m) compared with the TLS and SPM 5 m simulations (316 m and 469 m respectively). It had the least lateral spread and was concentrated in the middle of the valley, while the TLS and SPM better-captured the actual flow pattern. The total volume of snow in the release areas were more similar at 5 m compared with 2m as small topographic features begin to be smoothed, though the NZSoSDEM volume was lower (171,103 $m^3$) compared with the TLS (176,924 $m^3$) and SPM (177,128 $m^3$) simulations. The 5 m TLS simulation eroded the most snow (352,021 $m^3$) compared with the SPM (315,353 $m^3$) and NZSoSDEM (315,929 $m^3$). While none of the 5 m simulations matched the final debris pattern in the valley, the TLS had the greatest proportion of total debris volume (45%) located in the debris area. The SPM (32%) and NZSoSDEM (30%) had debris stop further down the valley. Similarities between the 5 m simulations include the core and powder velocities and powder pressures. However, the TLS simulation produced a larger estimated maximum core height (32 m) compared with the SPM and NZSoSDEM (both 19 m).

The 15 m simulations, also run on the TLS, SPM and NZSoSDEM surfaces, produced the most similar results (Figure 6, panel (d). All three simulations showed the core travelling too far down the valley with the bulk of the debris stopping


well-beyond the terminus rock. The maximum deposition and erosion was smaller owing to the spreading of the debris over a significantly larger area compared with higher-resolution simulations. Despite further smoothing of small topographic features with increased DSM resolution, differences remain in how the avalanches entrain snow into the core (total core volume for TLS: 310,952 m$^3$, SPM: 299,830 m$^3$, NZSoSDEM: 331,345 m$^3$). The total eroded volume followed the same pattern with the 15 m NZSoSDEM generating the largest core flow but converting the least mass into the powder cloud. None of the three

simulations characterized the final debris pattern in the valley with only 23% of the 15 m TLS debris volume located in the debris area, and only 20% and 30% for the SPM and NZSoSDEM respectively. While the maximum estimate core height was significantly greater between the TLS and SPM 2 m and 5 m simulations they were similar in the 15 m. The core and powder velocities and powder pressure were similar in all three surfaces. See Table 2 for summary of simulation outputs for each DSM and resolution.

We increased the depth of snowcover in the path, while keeping the average release depth the same, to mimic the avalanche flowing on a more snow-filled surface. Increasing the overall snowpack depth kept the release volume the same in the SPM simulation but actually decreased it marginally in the TLS (179,254 to 178,667 m$^3$) simulation showing the effect of resolving fine-scale terrain features. We found a similar pattern of difference between the TLS and SPM 2 m DSMs. Compared with the TLS simulation, the SPM had a lower maximum core flow height (28 m vs. 43 m) and lower core velocity (58 m s$^{-1}$ vs. 71

m s$^{-1}$). It nonetheless produced a larger avalanche by entraining more snow in the path (total core volume was 337,949 m$^3$ vs. 323,204 m$^3$ and total eroded volume was 481,545 m$^3$ vs. 439,686 m$^3$) and had a longer runout, traveling 201 m past the terminus rock, compared with 146 m for the TLS simulation. Interestingly, the runout distance was nominally the same for the SPM simulation run with and without the deeper snowpack, where the TLS simulation run with the deeper snowpack ran over 100 m further down the valley. In other words increasing the available snow in the path led to increased entrainment and total

core volume in both simulations. While the TLS simulation ran further down the valley, the SPM simulation did not, which suggests that decreasing the surface roughness of the sliding surface affected the TLS simulation more.

### 3.4   Topographic differences between DSMs

The DoD between the TLS-derived surface and the SPM-derived surface highlights important differences in the representation of terrain in the avalanche path. Since the denser TLS point cloud was registered to the SPM point cloud, overall the two

surfaces are well-aligned (Figure 8). The mean cell-to-cell difference for the entire 5.31 km$^2$ $TLS_{2m} - SPM_{2m}$ study domain was -0.13 m (RMSE = 4.25 m). The greatest differences are in areas of poor contrast in the SPM surface and in the cliff faces. For example, approximately 53% of the documented debris area in the valley was in shadow when the SPM satellite imagery was acquired. This area had a mean cell-to-cell difference of -0.7 m (RMSE = 4.17 m) compared with -0.1 m (RMSE = 0.64 m) in the remaining shadow-free portion of the debris area. The DoD results from steep terrain should be viewed with caution

as large errors can be created with cell-to-cell misalignment for the same terrain in two surfaces (Lague et al., 2013). Resolving sharp cliff edges in two DEMs, for example, can create large elevation differences.

    Nonetheless, subtle but consequential differences in the shape of the surface in gully features in the track and the delineation of cliff edges led to notable differences between the TLS and SPM surfaces. This is especially evident in the upper portion of

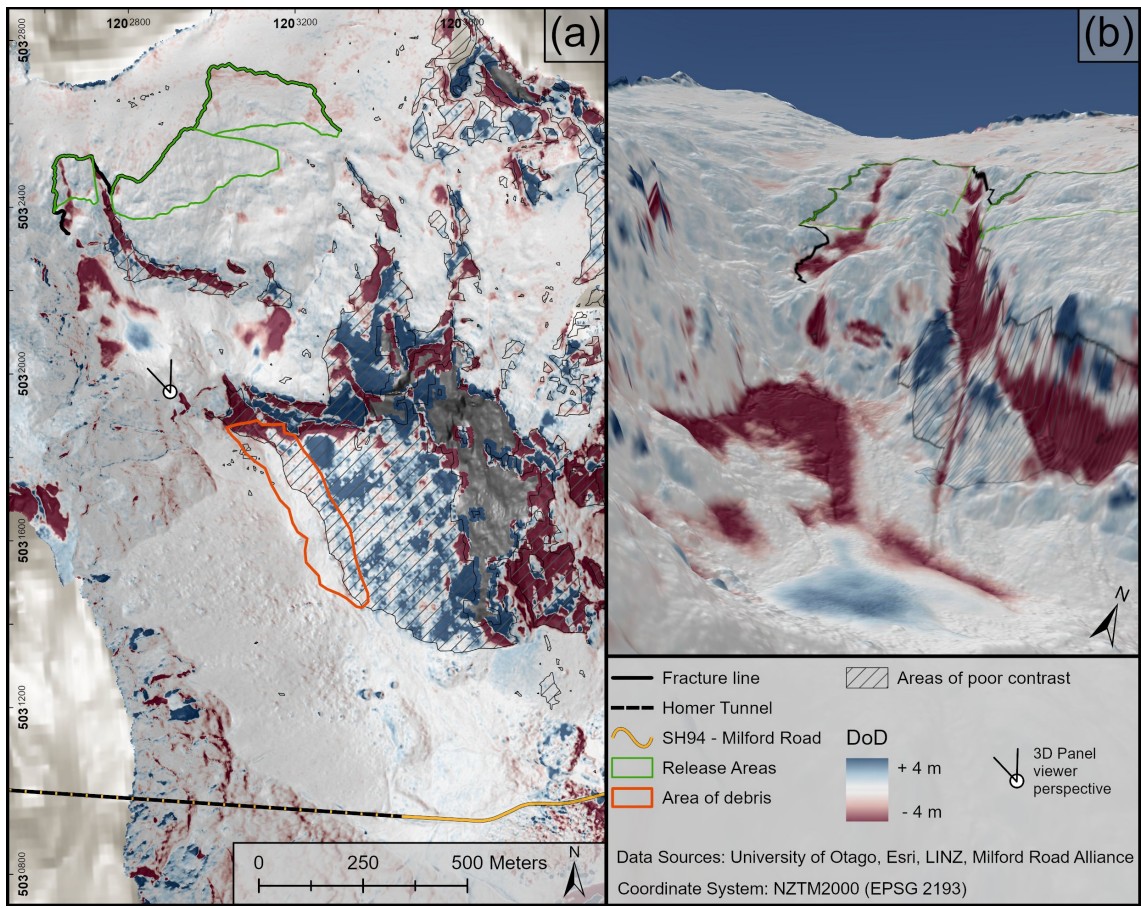

**Figure 8.** DEM of Difference (DoD) map showing topographic differences between TLS 2m and SPM 2m DSMs. The areas of poor contrast in SPM imagery are shown with hash polygon. The 3D panel focuses on differences in topography where snow melt occurred between data acquisitions and in the extent to which gully features are resolved in the DSMs.

the path where the orientation of gullies created poor view-angles to the satellite when imaged, and the true shape of the gully

features was not captured in the SPM DSM. The presence of seasonal snow on the bench between cliffs in the path also created clear differences in elevation between DSMs.

## 4   Discussion

After the McPherson avalanche was calibrated in RAMMS, the DSM data source and spatial resolution test revealed how differences in topography affected simulated avalanche behavior. Like Bühler et al. (2011), who compared DEM sources and

spatial resolution in RAMMS simulations, we found that increasingly coarse DSMs create longer core runouts. We also found powder cloud pressures and velocities varied considerably between finer- and coarser-resolution DSMs. At the same time,





subtle differences in terrain representation, based on the sensor technology and orientation to the terrain, also affected the simulated avalanche behavior between high-resolution 2 m DSMs generated from terrestrial laser scanning (TLS) and satellite photogrammetric mapping (SPM). In this section we will put some of the results in larger context, especially with regards to surface roughness and channelized terrain features. We will then discuss the implications for mass movement modelling posed by our study and make suggestions for hazard researchers and practitioners considering which elevation product to use in their modelling.

Testing differences in DSMs for snow avalanche modelling is a challenge because of the differences in the representation of above-ground features such as trees and bushes and their effect on measures of surface roughness. To avoid the influence of these features in the surface roughness, we simulated a snow avalanche that was flowing in a path lacking these above-ground features. The surface roughness was a better representation of the true sliding surface on which the avalanche ran.

## 4.1 Differences in surface roughness

Figure 9 compares the roughness between the 2 m TLS and SPM surfaces. Roughness was calculated as the vector ruggedness measure (VRM) (Sappington et al., 2007), which has performed well at characterizing roughness for avalanche modelling (Brožová et al., 2021; Bühler et al., 2018). We used a moving window area of 64 m$^2$ (4x4 cell neighborhood), to assess differences in roughness between the 2m DSMs. We also differenced the roughness maps to identify areas of diverging roughness. Overall the TLS-derived DSM has higher surface roughness throughout the path, the result of a higher point density in the point cloud before interpolation. Important differences are nonetheless still evident. The TLS is rougher in channel features in the bedrock located throughout the track. These channels are not fully resolved in the SPM-derived DSM, resulting in lower localized relief and thus lower surface roughness. Resolving these fine-scale terrain features in the TLS-derived surface meant more of the simulated avalanche core flowed through the channels in the upper track instead of spreading out, and likely increasing overall entrainment. If the Pléiades satellite had different orientation to the ground when the images were captured, or acquired as a tri-stereo image triplet, these features may have been more fully-resolved.

Differences in roughness is also evident in the runout, where part of the valley was in shadow at the time of the Pléiades image acquisition. The lower signal-to-noise ratio in the shadow promotes variability in stereo-matching and increases noise in the DSM (Eberhard et al., 2021), which in turn increases the apparent roughness. Since this was in the runout where the avalanche core had reached near-maximum velocity, the effect on the simulation results was reduced. However, the presence of shadow or other areas of low image contrast (e.g., saturated snow) should be considered when using photogrammetry-derived DSMs and can be mitigated depending on terrain orientation, region and date of imaging.

Differences in roughness in the release zones is also apparent, and the lower friction with the SPM surface may have increased both initial core velocity and entrainment. Differences in slope angle (Figure 9) also shows the smoothing of cliff edges in the SPM surface due to shadowing, large parallax and challenging geometry. Less crisp cliffs may have created conditions for the core to slide rather than eject over the cliff edge, potentially maintaining momentum for the core to travel further in the runout.



**Figure 9.** Composite map comparing slope angle and surface roughness for the TLS-derived 2 m DSM and satellite-derived (SPM) 2m DSMs. Panel (a) shows TLS 2 m slope angle, panel (b) shows SPM 2 m slope angle, and panel (c) is the SPM ortho image showing fracture line and debris area. Panel (d) is TLS 2 m surface roughness, panel (e) is the SPM 2 m surface roughness and panel (f) is the difference in surface roughness between TLS and SPM 2 m DSMs, which also shows the digitized release areas used in the simulation.

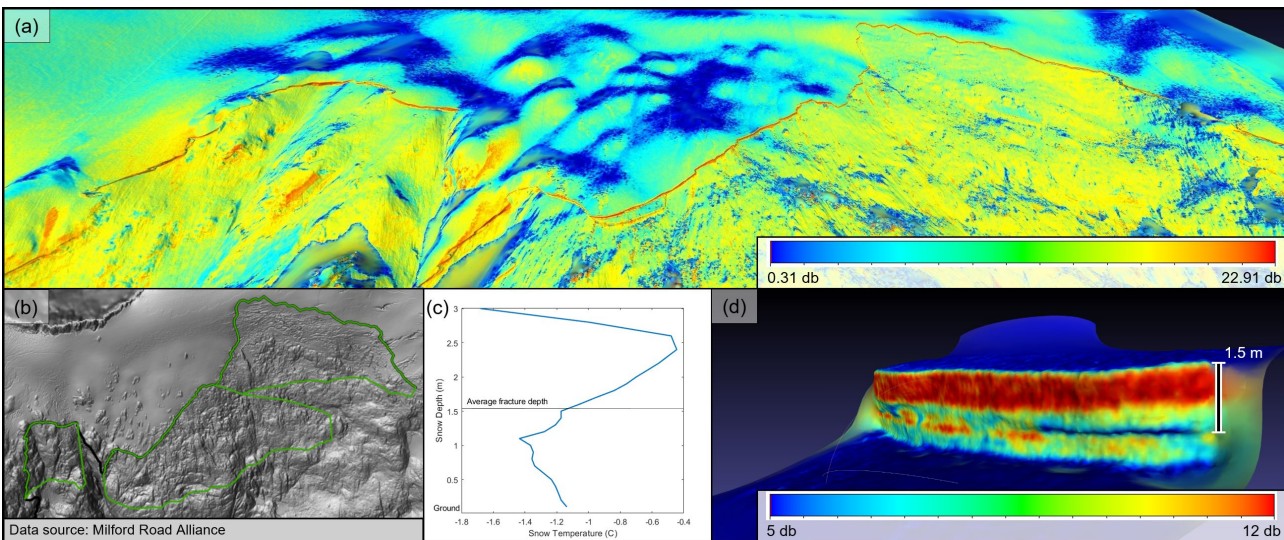

**Figure 10.** Composite showing amplitude of LiDAR return from scan of the release area the morning after the avalanche event in panel (a). Panel (b) is a hillshade of the release areas from the 0.5 m full-resolution TLS-derived DSM. Panel (c) shows the temperature profile of the snowpack at the closest weather station at 1700 m a.s.l. and depicting the average fracture depth used for the RAMMS simulations, based on the TLS data. Panel (d) shows a higher-resolution scan of the face of the crown wall from the morning after the avalanche, also visualizing the amplitude. Note the different scale bar ranges. The snow depth value in panel (d) is is indicative of the average fracture depth for the avalanche.

Figure 10 shows a scan of the release area the day after the event. By visualising the amplitude of the returning light energy to the sensor we can highlight differences in material – snow surface that did not avalanche, snow on the exposed surface after the avalanche and rock. The fracture line corresponded with the edge of the permanent snow seen in both the TLS and SPM surfaces. While the initial sliding of the avalanche slab may have been on a smoother snow surface than is represented by the DSMs, the avalanche removed most of the snowpack, followed by the avalanche tail depositing a shallow amount of snow on
the bedrock in the release area. A DSM with higher roughness created a more realistic surface for this event after the initial slab started sliding. Due to resolution limits, the smoother SPM DSM did not capture the fine-scale rough bedrock features captured in the TLS data after the avalanche. This was again evident when simulating the event with additional snow in the path to mimic a smoother winter surface. While the deeper snowpack resulted in more entrainment in the track, there was marginally more deposition in the release areas compared with the shallower (actual) snowpack depth. The lack of erosion suggests the rough
source surface still has an influence in the release dynamics. In the case of the McPherson avalanche, using a DSM – and one that better-resolves the true roughness of the terrain – improved the modelling. A newer generation of satellite sensors such as Pléiades Neo, which nearly halves the spatial resolution from Pléiades, will improve the terrain representation in rough alpine terrain lacking trees and shrubs.





Differences in how the shelf between cliffs in the track was resolved in the TLS and SPM 2 m DSMs also contributed to diverging core flow characteristics. More snow in the TLS simulation stopped on the shelf, which may partially explain the shorter (and more accurate) runout distance. The core reached the shelf first in the SPM simulation, which better-matched the timing from video evidence and again illustrates the smoother sliding surface in the release zone led to higher initial core velocities. Figure 11 shows the core as it flowed over the shelf at two time-steps in the simulation (t = 35 s and t = 40 s) for both the SPM and TLS 2 m DSMs. The smoother SPM surface meant the core travelled more directly down the track instead

of moving through the channels present in the terrain. Fine-scale channelling and protrusions from seasonal snow diverted the core flow in both simulations but to a greater extent in the TLS simulation. With increasingly coarse simulations, these features and become less influential. For example, in the 5 m TLS simulation some channelized flow is evident but with the 15 m TLS simulation the core flows straight over the shelf in a more homogeneous pattern. Slight differences in the distribution of seasonal snow on the bench at the time of the data acquisition for both the TLS and SPM surfaces is also evident.

**4.2 Effects of elevation source and spatial resolution on mass movement modelling**

These differences have implications for the ways in which model outputs are used (e.g., infrastructure design) but can also help clarify how certain topographic features may alter predicted avalanche behavior, which can be considered by the modeller when deciding on which elevation source and resolution to use. This study focused on an avalanche path without trees, where a DSM is a better representation of the actual terrain because of the rough nature of exposed bedrock as the sliding surface

throughout the track and runout zone. For terrain with high surface roughness, removing the first return in a LiDAR point cloud to create a DTM may create an over-smoothed surface for use in modelling. This may be appropriate in the release zone but not elsewhere in the path (Brožová et al., 2021), especially for smaller avalanches that may not break to the ground. The terrain and nature of this avalanche was such that the surface that better-captured the true roughness improved the dynamics in the release area as well. Our case may not be common to many sub-alpine modelling applications, especially where the study site

involves trees and shrubs. However it shows the value of using a DSM for many alpine cases.

Fundamental differences in the way terrain is represented can be traced back to the way it is captured. Photogrammetry – irrespective of platform – typically relies on two or more view angles from the sensor to the imaged surface, with a tension between the desire for high B/H ratio for matching accuracy and low incidence angles to avoid obstructions. Obstructions between an imaged area and the sensor by terrain in one image of a stereo-pair will create a data void or hole in the DSM.

Likewise, areas of cloud and areas of poor local contrast such as deep shadow or homogeneous snow, confound stereo matching and either increase noise or creates holes in the surface. These holes can be mitigated with increasing the number of images used in stereo restitution to achieve additional view-angles (e.g., tri-stereo instead of stereo, multiview stereo and structure-from-motion products) and/or using improved radiometric resolution suitable to the targets to be imaged. The presence of clouds cannot be mitigated and depending on the region may impact success of tasking, however increased temporal resolution

of imaging can increase the probability of cloud-free images.

Interpolators can be used to fill holes, but the size of the hole to be filled should be a considered carefully and the relevance assessed based on predominant aspects and where holes are located in the simulation domain. For example, in this case most


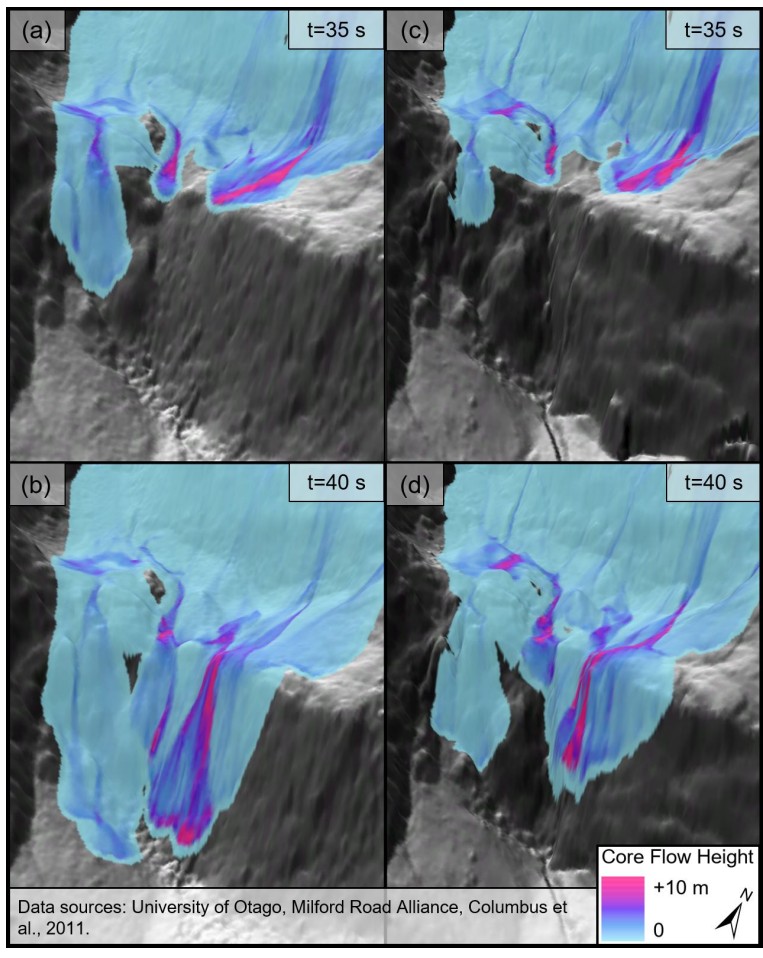

**Figure 11.** Composite map showing the estimated core flow height at two time steps for both the TLS and SPM 2 m DSMs. Panel (a) shows the SPM DSM and panel (c) shows the TLS DSM both at t = 35 s in the simulation. Panel (b) shows the SPM DSM and panel (d) the TLS DSM at t = 40 s in the simulation. For spatial context, the cliff face shown in the composite is approximately 200 m in height.

of the large holes were located in terrain where the simulation was unlikely to encounter them on steep walls adjacent to the path. These holes were not filled based on our threshold of 100 m$^2$ contiguous area. Working with lower-resolution DEMs will

decrease the proportion of the domain covered by holes as smaller holes will be filled with standard interpolation. Thought should also be given to how the interpolation of holes is likely to create an unrealistically smooth representation of terrain, which, as discussed, has implications for how a simulated mass movement flows over the terrain.

LiDAR can mitigate the presence of holes in the DEM as areas of poor contrast will not meaningfully impact the measurements. However, holes from obstructed view of the terrain remain. TLS-derived DEMs are more likely to have holes given the

oblique scan angle to the terrain, but can be mitigated with multiple scans used to build a composite. Manual cleaning of the point cloud is likely needed to avoid artefacts. Local sinks and spikes can create spurious simulation results. This is particularly





important in terrain with extreme roughness (jagged cliffs, crevasses, overhangs) where points can be misappropriated to an output cell in the interpolation. The difference between a summer and winter surface is especially pronounced in glaciated terrain where an avalanche will flow over crevasses filled with snow in winter by represented as deep slope-perpendicular
channels in the summer DSM. Filtering may be necessary to fill these features for use in avalanche modelling.

    The wide range of terrain considerations means there should not be a universally recommended simulation resolution for hazard models (Bühler et al., 2011; Claessens et al., 2005). The rate of landscape change in many regions means the elevation data my not reflect the true terrain if captured some time before the modelling. At the same time, previous avalanching may create a different surface for the avalanche to run on than is what is represented by the DEM. Generally, these factors become
less important with coarser DEMs. We found that the 15 m DSMs produced the most similar simulations since the influence of fine-scale terrain features is dampened or removed. However, a loss of permanent snow in the upper portion of the path since the 1988 aerial imagery used to generate the NZSoSDEM created steeper, rougher terrain evident in the TLS and SPM 15 m DSMs, and altering the core flow characteristics in the upper portion of the track. Down-sampling the 15 m NZSoSDEM to 5 m also created artefacts in the simulation with a significant deposition of snow in the lower part of one release area. The age of
the DEM, especially in alpine regions, should be a consideration for whether the DEM is appropriate to use for modelling.

    Overall the coarser DSMs poorly captured the characteristics of the McPherson avalanche, with implications for design specifications for infrastructure or operational decision-making. For example, estimated impact pressures varied considerably between simulations and were largest in the highest-resolution DSMs where channelized terrain features were resolved, compared with the coarser DSMs where the core spread out more. The most accurate estimates for roading infrastructure design
would need to account for these subtle terrain features. Overall, we found that the use of a coarse DEM for avalanche modelling in steep, rough alpine terrain is not appropriate. However this is often the only DEM product available to modellers in many alpine regions. Up-sampling the coarse DEM to a higher resolution will not improve the representation of terrain nor the model. At the same time, the use of a very high-resolution (< 2 m) DEM not only drastically increases the model processing time but also does not improve the results. We found the 1 m TLS-derived DSM simulation had too much friction and the simulated
avalanche failed to replicate the behavior of the documented avalanche. This is most important in the upper path where the finely-resolved features were too rough for the avalanche to gain momentum appropriately.

    As the number of high-resolution DEMs in alpine regions increases, absolute accuracy of each DEM will become more important. Currently, the patchwork of elevation products in many areas means it can be challenging to assess the accuracy of a project-based DEM against a reference DEM. This is not an issue with many modelling applications where a DEM with
high relative accuracy is sufficient. However, with many disparate DEMs produced through time, absolute accuracy becomes important to quantify landscape change, detect artefacts in the DEM and tie in with other contextual data, related to, for example, infrastructure or land-use.

    Aerial LiDAR, not available for our study site, would allow for additional sensitivity testing given the orientation difference to the sensor and the expected completeness in the output DEM. That said, some of the jagged and overhanging cliff faces are
better captured from the ground. An advantage of the TLS is also rapid deployment after an event to refine model parameters and provide snowpack distribution information for forecasters (Prokop et al., 2015; Thibert et al., 2015; Deems et al., 2015;


Maggioni et al., 2013; Prokop, 2008; Sailer et al., 2008) as an alternative to more costly aerial LiDAR (e.g., Sovilla et al. 2010). In our case the scan of the release zone from the following morning provided important information on the characteristics of the avalanche. Remote sensing techniques were the only viable option to estimate the release depth and identify where in the
snowpack the weak layer was located. Figure 10 shows how remotely measuring the crown wall and correlating the weather station data on snow temperature and density was used for precise model parameterization. The slab that released is discernible in the point cloud, owing to the sensitivity of the sensor, which was up to 1,800 m away from the crown wall. The time between the avalanche and scan (approximately 18 hours) is a limitation to further investigation of the reflectivity of the snow in the crown wall and release zone in this case, but nonetheless demonstrates the utility of rapid deployment of the TLS for avalanche
model calibration.

Another limitation of our study was the slight difference in timing for the avalanche to reach the valley between the calibrated simulation and the video evidence. This two-second difference over 39 total seconds may be partially reflective of the challenge of modelling the plunging dynamic of the avalanche. While the coarser DSMs produced higher initial velocities in the upper track, the slightly slower calibrated avalanche much more accurately captured the flow patterns lower in the path, the behavior
of the powder cloud and the debris in the runout. Despite the conservative core velocity estimates, the avalanche detached from the lower cliff at a high velocity (in excess of 60 m s$^{-1}$) and splashed across the terrain in the valley, creating an unusually large powder cloud from a warm snowpack. RAMMS performed well to replicate the challenging behavior of a large avalanche on steep terrain in a maritime climate.

## 5 Conclusions

We used high-resolution digital surface models (DSMs) to test the sensitivity of RAMMS snow avalanche simulations for elevation source and surface resolution. Building on the sensitivity test by Bühler et al. (2011), we investigated current state-of-the-art sensors and platforms. The simulation using the 2 m DSM derived from terrestrial LiDAR best represented the terrain complexities in the steep avalanche path, compared with finer (1 m) and coarser DSM (5 m, 15 m) resolutions. Increasingly coarse DSMs produced longer core runouts, entrained more snow and yet produced lower estimates of core flow heights and
powder pressure. The implication of this finding is that hazard modellers should be cautious when using coarse DEMs for avalanche simulations in steep, rough terrain. We also found that subtle differences in the representation of terrain features like gullies in high-resolution DSMs from different sensor technologies (terrestrial LiDAR and satellite photogrammetry) also influenced simulated avalanche behavior. There are three main lessons from this study that apply to snow avalanche modellers, as well as hazard modellers more broadly.

1. A high-resolution DEM is necessary for modelling snow avalanches in complex terrain. Starting with a higher-resolution DEM (e.g. 1 m) and up-sampling to a coarser DEM for modelling efficiency is appropriate. If the terrain has gully or channel features, especially for smaller avalanches, a high-resolution DEM is especially important.

    2. The use of a DSM in hazard modelling can be appropriate to some terrain settings, thus confirming findings by Brožová et al. (2021). A DTM may artificially smooth topographic features, but at the same time, a DSM may not best reflect the



initial sliding conditions of a slab avalanche in the release zone. The size of the avalanche and whether the modelling is
occurring in alpine terrain or sub-alpine terrain, also have implications for the sensitivity of the simulation to the DEM
source and resolution. Future research is needed on how best to optimize a DEM for local topography and vegetation.
For example, a dynamic DEM that mimics snow-on conditions in the release area from a smoother or coarser resolution
surface and snow-off conditions in the track and runout with a rougher or finer resolution surface may better-capture true
sliding conditions in the path.

3. High-quality elevation products are available from a variety of platforms (terrestrial, RPAS, aerial, satellite) and technologies (photogrammetry, LiDAR, InSAR), each with advantages and disadvantages. Satellite photogrammetric mapping (SPM) provides a relatively affordable way to generate an accurate high-resolution DSM over a large geographic (>400 km$^2$) area. Satellites can be tasked quickly after an event and image processing pipelines can deliver a relatively accurate DSM in hours. The presence of clouds and deep shadow must be considered when tasking the satellite, however.
The unfortunate presence of clouds in the study domain will create data holes that prevent accurate modelling. Areas of poor contrast such as shadows or fully-illuminated homogeneous snow cover increase the measurement uncertainty and could also create holes in the DSM. Small holes can be successfully filled with standard interpolation techniques, but interpolating large areas will misrepresent the terrain with implications for modelling. While bi-stereo 0.5 m resolution satellite imagery processed in this study showed the capabilities of the product for hazard modelling in complex topography, tri-stereo acquisition is advised in such terrain. Higher-resolution imagery from the next generation of satellite sensors (e.g., Pléiades Neo with shortened revist times) is expected to improve DSM availability and suitability for mass movement modelling.

Terrestrial laser scanning (TLS) offers a more detailed model of the terrain. It better-captures subtle terrain feature shapes
such as channels, especially in regions with vegetation, which improves the hazard modelling. However, more data processing and manual cleaning of the point cloud is necessary to generate the DEM. At the same time, it may take multiple scans to completely cover the study domain. The advantage of TLS over aerial laser scanning (ALS) is the rapid deployment after an event to document landscape change, as the expense, logistics and data processing of ALS surveys mean they occur relatively infrequently and are not available in many regions. Also, the rate of landscape change means elevation products may have data
relevancy issues. The rapid deployment potential of TLS and SPM improve data relevancy.

Finally, RAMMS performed well to simulate the characteristics of the McPherson avalanche. In addition to the terrain complexities, the snow conditions typical for Fiordland avalanches required precise model calibration to generate an appropriately sized powder cloud from a warm snowpack with a steep snow depth gradient through the track. Further research on the dynamics of wet avalanches capable of generating powder clouds will become increasingly important as many regions are reporting
a shift towards warmer avalanches.



*Author contributions.* AM,PS and YB conceived of the study. AM,SM,PS,TR and KT contributed to data collection. AM and PS processed the satellite imagery. SM and AM processed the LiDAR point clouds. AM and PB performed the RAMMS simulations with input from YB. AM,SM and TR prepared the manuscript with contributions from all co-authors. PS, NC and YB supervised the research.

*Competing interests.* Some authors are members of the editorial board of NHESS. The peer-review process was guided by an independent
editor, and the authors have also no other competing interests to declare.

*Acknowledgements.* This research was supported by New Zealand Ministry for Business, Innovation & Enterprise through the Endeavour Smart Idea research project "Quantifying environmental resources through high-resolution, automated, satellite mapping of landscape change" (grant no. UOOX1914). We would like to thank Downer NZ Ltd. Milford Road Alliance for logistical support for fieldwork and providing the LiDAR data. We would like to thank Etienne Berthier at LEGOS and the French space agency (CNES) for the use of the satel-
lite imagery made available through the Pléiades Glacier Observatory (PGO) initiative. Contains information © CNES (2020), and Airbus DS (2020), all rights reserved. Commercial uses forbidden.



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
