# Peer review of "The impact of terrain model source and resolution on snow avalanche modelling"

_Natural Hazards and Earth System Sciences, 2022_

## Author Comment (AC1)

**Anonymous Referee #1 (RC1)**

Dear Anonymous Referee #1

Thank you for your constructive review of our paper.

We have provided answers to your questions and suggestions (your comments in bulleted italics) below:

*1) general comments*

- *The manuscript investigates the influence of selected digital elevation models from different sources and resolutions on the modeling of a snow avalanche. Model runs and model output parameters are compared to a reference avalanche event. The manuscript could be improved by providing more information about the practical usability of the different model results and the transferability of findings.*

We will look at extending the Discussion to provide more information about both the practical usability of model results and transferability of findings to other regions with different topographic settings.

*2) specific comments*

- *Abstract: Please note that digital elevation models are always a 2.5 D representation and not fully 3D.*

We will update the Abstract to introduce the concept of a DEM as 2.5D representation of terrain used to represent true 3D terrain in a computer system.

- *Abstract: Please use the term "topographic LiDAR" throughout the whole paper.*

We will use the term "topographic LiDAR" as an umbrella for the platform-specific acquisition mapping techniques (e.g. terrestrial laser scanning).

- *Abstract: "performed well" - please add quantitative results in the abstract as well.*

We will update the abstract to include the quantitative results from the 2 m TLS simulation.

- *Line 189: Sections 1.4.2 climatic setting and 1.4.2 avalanche mitigation are not relevant for a study analyzing the effect of DEM resolution. Please delete or shorten.*

We appreciate this point about the focus of DEM resolution. We believe the climatic section (1.4.2) gives important context to how topography and snowpack conditions combine to produce wet avalanches that generate powder clouds and run into a snow-free valley, which the RAMMS modelling needed to effectively simulate. This combination of snowpack and terrain may be limited to maritime regions, however, the setting provides a case for how well RAMMS performed when using a high-resolution DSM, which we believe is of interest to readers.

We will delete the section 1.4.3 on avalanche mitigation.

- *Line 213: Please add a workflow diagram visualizing all processing and analysis steps providing a better overview for readers.*

Thank you for this good suggestion. We will create a new figure that details the workflow for the data processing and analysis.

- *Line 219: What is the motivation to make use of the selected DEM sources? There are several other options, which would be also interesting for comparison e.g. why not making use of Pleiades tri-stereo, ASTER GDEM, etc. Please explain to the reader what DEMs are available for the specific site investigated and in New Zealand in general.*

We used the best available DEMs for the study site in terms of spatial resolution and quality. While tri-stereo Pléiades would have been preferred given the high topographic relief in the study site, it was not available. Prior to our satellite image and terrestrial LiDAR data collection, the best DEM available was the NZSoSDEM (also used in the study). This is currently the most accurate DEM covering the entirety of Aotearoa New Zealand, even though it was produced from 20 m contours generated from aerial photogrammetry in the 1980s. Modern aerial LiDAR and both aerial and satellite photogrammetry datasets are available as a patchwork across the country. Some geographic areas in Aotearoa New Zealand have numerous modern high-resolution datasets available, but these are clustered in populated regions. Much of the mountainous and remote regions of the country still lack high spatial resolution DEMs.

Global DEMs were not the focus of this study as Bühler et al. (2011) already conducted a similar sensitivity test using coarser global DEMs, including ASTER and SRTM and found artefacts present in the global DEMs created artefacts in the RAMMS simulations. Caution was urged in using the coarse DEMs, especially in terrain with high surface roughness. Section 1.2 details the state of DEMs in Aotearoa New Zealand and globally.

We will revisit this section in the paper to briefly explain what the available DEMs were in our study site and refer the reader back to Section 1.2 for further information on DEMs elsewhere in Aotearoa New Zealand and globally.

- *Line 237: Please add information about the resampling method and settings used instead of only naming the tool.*

There is an important distinction between point cloud interpolation and resampling. With the exception of the downsampled 5 m NZSoSDEM, none of the other DSMs used in the study were resampled. Rather, they were interpolated directly from the source point cloud (TLS, SPM, NZSoSDEM) to leverage the higher point density in the interpolation of the coarser DSMs generated from the source point cloud.

The *point2dem* tool (Beyer et al., 2019; specific tool documentation: https://stereopipeline.readthedocs.io/en/latest/tools/point2dem.html) uses a search radius set by the user to calculate a weighted average of all points with the weights given by a Gaussian to apply to the output cell in the DSM. The search radius is a function of the cell size of the interpolated grid. In our case we used a search radius of 1.2, so the DSM generated at 2 m

from the SPM point cloud, for example, included a sample of points within a search radius of 2.4 m when applying the weighted averaging. We discuss the use of the *point2dem* tool from ASP in Section 2.1.4 on hole filling (Line 286), but we will update this section to include an account of the interpolation method.

- *Line 251: Can you demonstrate the change in roughness without and with cleaning? How is this step reproducible or transferable to other sites?*

The upper path includes permanent snow that was covered in seasonal snow in some individual scans. In creating the composite point cloud, the points associated with the seasonal snow were manually removed using the Riegl's RiSCAN Pro v2.14 software. The point density of the point clouds made manual cleaning efficient as each individual scan was discernible, aided by visualising the reflectivity of the surface and with an approximate vertical offset of 1 m from the points least affected by seasonal snow. Some seasonal snow was still present in the upper path above the fracture line of the avalanche, which remained in the composite scan. We will update this section on the extent of manual cleaning necessary when combining multiple scans into a TLS composite point cloud.

The implications for a rougher initial sliding surface as represented by the DSM are expanded on in the Discussion section (lines 535-543). Including seasonal snow in the composite cloud would have made for a less comparable "summer" DSM alongside the satellite photogrammetric mapping DSM.

- *Line 253: "other times of year": Does the phenological stage of vegetation have any influence on the surface representation? You may demonstrate the effect at overlapping areas.*

The vegetation in the study site is not deciduous with similar structure in summer and winter. The only vegetation in the path itself is alpine grasses and sparse shrubs in the runout (see Figure 2 for example). In the case of the avalanche event in this study, the final deposition occurred up valley from any shrubs. While this is good for testing the DSM sensitivity, we recognize it will not be the case in many other regions. We will update this section to comment on the importance of phenological timing of TLS acquisition when using multiple scans in a composite.

- *Line 259: "without coordinate transformation" - Please specify. What coordinate system would be used in this case? How can you be sure that your point cloud model is oriented along with the horizontal and vertical axis correctly?*

The composite point cloud was generated in a project coordinate system in the Riegl RiSCAN Pro v2.14 software, where the 0,0,0 (*X,Y,Z*) origin was located at the center of the scanner from the main scan location approximately 50 m east of the Homer Tunnel Entrance (See Figure 1). The horizontal reference was based on magnetic north, and the vertical reference was the base of the internal laser plumb level.

The composite point cloud was exported as a LAS file with coordinates in the project coordinate system. The point cloud was then manually aligned to the SPM point cloud initially using CloudCompare v 2.10.2 before co-registration with ICP and retention of the transformed coordinates in an absolute referencing system, New Zealand Transverse Mercator, EPSG 2193 (Lines 267-268).

We will update this section to specify the project coordinate system origin and horizontal and vertical axis orientation.

- *Table 1: Please specify acquisition dates for TLS scans.*

We will update Table 1 with the acquisition dates for the TLS scans.

- *Line 284: "interpolation" – please specify the interpolation strategy and settings used and document the influence on DEM quality.*

As discussed in the comment above on resampling, the interpolation was performed with ASP's *point2dem*, which uses a search radius set by the user to calculate a weighted average of all points with the weights given by a Gaussian to apply to the output cell in the DSM. The search radius is a based on the cell size in the output DSM grid. This interpolation method for creating a DSM from a point cloud has been applied to a number of sensors and terrain types (Sykes et al., 2021; Eberhard et al., 2021; Deschamps-Berger et al., 2020; Beyer et al., 2018; Shean et al., 2016). To produce the "full-resolution" DSM from the stereo satellite imagery, we followed the same method as Eberhard et al. (2021) and Shean et al. (2016) and used a 4x factor to produce a 2 m gridded DSM from the 0.5 m panchromatic imagery. Shean et al. (2016) found improvements in the DSM quality with the 4x grid spacing in terms of noise reduction and artefact mitigation.

When producing the coarser DSMs, we used the same full-resolution point cloud while adjusting the output cell size (e.g. 5 m, 15 m). For consistency, we produced the TLS and NZSoSDEM DSMs with the same method to limit any differences in topographic representation from the interpolation method.

- *Line 292: Please provide quantitative information on "required less hole-filling".*

Reviewer #2 also asked for clarification around the hole-filling process. We realise that our statement of "required less hole-filling" may have been confusing. Hole-filling is a function of the cell size so there were a greater number of cells filled in the finer resolution DSMs compared with the coarser DSMs. While the full-resolution TLS DSM (0.5 m) had the greatest proportion of the DSM comprised of holes (3% of the RAMMS modelling domain) the coarser DSMs had a smaller proportion of the area comprised of holes (e.g., 0.05% for 5 m DSM). We will update this section to discuss the reasons for hole-filling and clarify the workflow.

- *Line 306: Is the code of your developed script somewhere available for the scientific community?*

The script is part of an operational toolkit used by the Milford Road Alliance. Due to commercial sensitivities, it unfortunately cannot be shared publicly at this time.

- *Line 327: Please quantify "better represent the true terrain".*

As discussed in comments on interpolation and resampling above, we interpolated the DSMs of varying spatial resolution (1 m, 2 m, 5 m, 15 m for TLS, 2 m, 5 m, 15 m for SPM and 15 m for NZSoSDEM) directly from the full-resolution source point cloud. This approach leverages a greater number of points used in the interpolation of the coarser DSMs. The

typical approach of resampling from a higher resolution DSM to lower resolution DSM relies on fewer points in estimating an output cell elevation value and a less smooth transition between cell elevation values. We did downsample the NZSoSDEM from 15 m to 5 m to demonstrate the sub-optimal resultant DSM quality and artefacts when downsampling a DEM and to provide another 5 m RAMMS simulation for comparison.

We believe an additional sensitivity test on the interpolation from the source point cloud compared with resampling the grids themselves to quantify the differences would not add substantively to the focus of the paper. We will therefore remove this clause so the sentence will read "This means that a greater number of elevation measurements are used to interpolate coarser DSMs ."

- *Line 331: Please explain and motivate your decision using two snow layers in the model. How did the weather station data look like so that you decided for a two layer setting?*

We tested both one and two snow layers in the RAMMS simulations. We found the two layer model performed better in terms of final deposition pattern in the runout. We used a top layer depth of 1.5 m based on the average fracture depth (1.53 m) and a bottom layer depth of 1.5 m based on the remaining snowpack at the weather station located adjacent to the path (Figure 1). The two-layer simulation allowed us to better-reflect the temperature profile of the snowpack (Figure 10) from the weather station where the upper portion of the snowpack was warmer than then lower portion. We calculated the mean snow temperature for both the upper 1.5 m and lower 1.5 m of the snowpack based on temperature values collected on 0.1 m intervals at the weather station. The mean temperature of the top layer was -0.8 °C and the bottom layer had a mean temperature of -1.3 °C, which were used in the simulations.

We will update this section to explain the use of the two-layer simulation.

- *Line 344: Please list all model parameters and input data sets used in different test runs, e.g. in a table as an appendix.*

We will add a table in the appendix/supplementary.

- *Line 360: Please add values for co-registration quality.*

We report on the results of the DoD in the Results section (Section 3.4, lines 478-486; Figure 8) which we believe is the appropriate place for the co-registration quality values, rather than in the Methods section.

- *Line 376: You estimate a reference volume derived from TLS and PlanetScope. Can you add information on how safe this value is considering the resolution and uncertainty of input data sets and preprocessing steps? What deviation would you allow from this reference value interpreting a model result still as correct?*

Thank you for raising this important point. Our reference volume is based on the results of the 2m TLS simulation and not a direct measurement. The oblique angle from the TLS point cloud of the debris following the avalanche prevented a direct estimation of the deposited volume. Nevertheless, we used the scan to corroborate the debris area observed in the

PlanetScope image (3 m resolution), although some uncertainty remains in the delineation of the debris area. To partially account for this we compared debris volumes both in our delineated debris area, as well as the total debris below the second cliff (labelled as "Total deposition volume in runout", Table 2). The reference simulation had 99% of the total debris volume in the runout located inside the debris area manually delineated by the TLS and PlanetScope image, though the tow of the debris did extend 28 m further than the documented toe. Given the size of the avalanche and the terrain, as well as extensive expert input and critical analysis involved in generating this output, we consider this a well-calibrated simulation, which provided a baseline to compare against other simulations where only the DSM source and resolution were varied. The debris volume remains a modelled result nonetheless.

The goal of this study was to test the sensitivity of simulations to DSM source and resolution, based on a well-calibrated reference simulation. In effect we are comparing a model result to another model result, which we believe offers important insights into the model sensitivity to terrain representation when interpreting the results. Numerical hazard modelling still requires expert knowledge of the snowpack and terrain—for model calibration, interpretation of the results and when deciding how to inform operational use of the results. Quantifying acceptable/unacceptable deviation from the reference value still requires expert input, critical thinking, and arguably some degree of subjectivity, unless an unambiguous and fit-for-purpose metric is defined against which "acceptability" is measured for a particular outcome (e.g., peak impact pressure at some location). This would require a specific methodology for example generating a large ensemble of simulations to produce statistically relevant distributions of outputs. Although we can only agree this would be a valuable endeavour, it is unfortunately not within the scope of this study.

In our case, the main objective observations are limited to the runout length and the location of the deposited debris. From an expert, yet admittedly subjective approach, we believe it is enough to assess with some degree the unrealistic and limited credibility of some simulations. The use of a set of reference outputs derived from the calibrated simulation adds to the comparison by demonstrating which outputs are more sensitive to the representation of terrain in the dynamic hazard model. Nonetheless, we do not believe the current methodology provides enough statistical strength to unambiguously quantify what deviation in model outputs should be judged unacceptable.

- *Line 378: What do we learn from the mass values for the tests performed in this study? Are they relevant?*

The mass of an avalanche has strong influence on the friction parameters and therefore on velocity, runout distance and the impact pressure, also for the powder cloud. Avalanche mass is an important variable taken into account in more simplified avalanche models such as the RAMMS::AVALANCHE user version (Christen et al., 2010), SAMOS (Sampl and Zwinger, 2004) or r.avaflow (Mergili et al., 2017). From an operational perspective, avalanche mass estimates are useful for comparing avalanches through time. The Milford Road Alliance, who manage the avalanche hazard along the highway running through the study site, keep detailed records of avalanche events, including estimated mass, release area, fracture depth, runout length, and impacts to roading infrastructure. These records are used to help refine a hazard index for each path affecting the road. Estimated mass is an important contributor to the index. The RAMMS simulations offer another estimate of mass to compare to their existing methods to inform operational decision-making.

At the same time, the estimated mass values from the tests performed in this study demonstrated how changing the spatial resolution of a DSM altered the estimated core mass. For example, the total core mass estimated by the RAMMS simulation on the 15 m TLS DSM was 14% greater than the 2 m TLS simulation, suggesting a coarser DSM will increase simulated entrainment within the path. Like the other event metrics (maximum core and powder pressure, runout length, etc.) topographic representation affects the simulation outputs. However, using the same DSM source and resolution, supports the comparison of different avalanche events and their implications for future hazard management.

- *Line 379: Please explain avalanche classification "size 5" in more detail.*

Comparing avalanches by size is a challenge for a number of reasons. For example, variations in snowpack (cold, dry, warm, wet) release and flow conditions (slab, loose) and entrainment result in different combinations of mass, runout lengths and destructive potential from the core and powder cloud. Nonetheless, a single classification system for avalanche size and destructive potential for event comparison is operationally desirable. We use Size 5 here based on the Canadian avalanche classification system (McClung and Schearer, 1980) and later refined and adopted elsewhere (Moner et al., 2013). Based on this classification the largest avalanches are classified as Size 5 where the "typical length" is greater than 3 km and the "typical volume" is greater than 100,000 $m^3$. Our reference simulation had a release volume of 179,254 $m^3$ and total core volume of 267,610 $m^3$. We will update this sentence with the criteria for the Size 5 classification.

- *Line 380: Please link avalanche properties mentioned here to the results in Table 2.*

We will update the sentence on results from the calibration with a link to Table 2.

- *Fig. 5: Please add elevation profile lines in the two maps (subfigures upper and lower right).*

Thanks for this good suggestion. We will update the figure to include elevation profiles on the maps.

- *Line 388: Here authors explain the special nature of the topographic situation of the site investigated. Are the results of the study transferable to other topographic situations? Can you please add information on the transferability and generalization of the findings in this study?*

By using a site with high relief and alpine features (no trees being the most distinguishable feature) we are limiting the direct comparisons to some other sites. However, this test shows the importance of the underlying terrain representation in hazard modelling and could be extended to other regions with differing topographic situations. As the prevalence of high-resolution DSMs increases with a variety of platforms and technologies (e.g., TLS, RPAS and structure-from-motion), these DSMs will inevitably be used for modelling applications. Our results show that caution should be used when deciding on what surface to use in dynamic modelling and when interpreting the results.

We will update the discussion and conclusion to expand on the transferability and generalization of the findings.

- *Line 429: Differences in simulations results are described in detail. Can you add information at which order of magnitude differences in simulation results become relevant i.e. model results become insufficient for hazard management applications?*

There is no standard guideline on how to apply model results for practical hazard management applications, but rather this is based on the interpretation of avalanche experts. We report the details of the simulation results for readers to be able to compare to other existing and future modelling results. Like in Bühler et al. (2011) we tested a range of elevation sources and resolutions, however unlike with their test we changed the RAMMS simulation resolution to match that of the input DEM. At the same time, we modelled a large avalanche in rough terrain, which showed that even with a large avalanche, the use of a high-resolution DSM was necessary. The most salient model results when distinguishing one simulation from another are the runout distance (operationalised as the furthest reach of the debris past a large rock where the observed debris stopped) and the deposition volume located in the debris area relative to the 2m TLS reference simulation.

The order of magnitude simulation results become relevant depends on how the results will be used (e.g., hazard mapping, safety assessments, longer-term inventories). In all cases, expert interpretation of results (see comment on model result interpretation above) is necessary. For our study specifically, we used the coefficient of variation (CV) in Table 2 as a way to show how clustered or dispersed the model's results were from the mean. This is not a perfect metric as the mean values are less meaningful than the reference values, however it does highlight the greatest divergence among model results. From this we can see the runout distance had the greatest dispersion of the model outputs (CV of 87%). The second greatest dispersion came from the proportion of the total debris in the runout that was located in the documented debris area (CV of 54%). The best agreement with the 2 m TLS reference simulation through all the model results was with the 2 m SPM simulation. We found that in our study site, we would caution against using the 5 m or coarser simulations, while the 2 m TLS simulation effectively characterised the documented event. Future work could focus on identifying the most suitable resolution given the terrain in the modelling domain.

We will update the results section to highlight the two most salient model results (runout and deposition volume) when interpreting the results.

- *Fig. 7: Please add elevation profiles of DEMs.*

Thanks for this good suggestion. We will update the figure to include elevation profiles alongside the surface difference profiles.

- *Line 487: Please specify. What is the order of magnitude for gully features to be relevant in avalanche modeling?*

This will depend on the size of the avalanche. In this study, gulley features in the order of 5-10 m deep and 10-20 m across were found to divert the core flow of the avalanche. Gulley features existed in both the TLS and SPM DSMs, however their shape and definition differed, as revealed by the DSM of difference (DoD). Figure 8, panel (b) exemplifies the differences in gulley representation between the DSMs. As an example, we show in the figure below a cross-slope profile (checked line) of the 2 m TLS and 2 m SPM DSMs across a gulley feature where poor view-angles to the satellite in the SPM surface failed to capture the depth and shape of the gulley. Figure 11 shows the core flow height at two time-steps in the simulation

where the presence of smaller gulley features (on order of 1-5 m deep, 5-10 m across) were affecting the flow of the avalanche on the bench between cliffs, which resulted in more snow stopping on the bench in the TLS simulation, compared with the SPM.

[Figure]

- *Line 506: Please rephrase this sentence to be more clear.*

We agree this could be clearer and this sentence is unnecessary. We will delete the following sentence

- *Fig. 10: Please add an overview map with marked areas of shown subfigures for readers, who are not familiar with the test site.*

We will update the figure to include an overview map with the subfigures identified.

**# technical corrections**

- *Line 35: cartesian coordinate system*

We will use the lower-case "cartesian coordinate system" to reflect the American spelling style adopted by the paper.

- *Figure 3: Please use a different color for the fracture line for better visibility.*

We will update Figure 3 with a colored fracture line.

- *Fig. 9: Please add a blank between numbers and SI units.*

Thanks for catching this. We will update Figure 9 with a space between value and SI unit.

**References**

Beyer, R. A., Alexandrov, O., & McMichael, S. (2018). The Ames Stereo Pipeline: NASA's open source software for deriving and processing terrain data. Earth and Space Science, 5(9), 537-548.

Beyer, R., Alexandrov, O., and McMichael, S.: NeoGeographyToolkit/StereoPipeline: Ames Stereo Pipeline, v2.7, https://doi.org/DOI:10.5281/zenodo.3247734, 2019.

Bühler, Y., Christen, M., Kowalski, J., & Bartelt, P. (2011). Sensitivity of snow avalanche simulations to digital elevation model quality and resolution. Annals of Glaciology, 52(58), 72-80. doi:10.3189/172756411797252121.

Christen, M., Kowalski, J., & Bartelt, P. (2010). RAMMS: Numerical simulation of dense snow avalanches in three-dimensional terrain. Cold Regions Science and Technology, 63(1-2), 1-14.

Deschamps-Berger, C., Gascoin, S., Berthier, E., Deems, J., Gutmann, E., Dehecq, A., Shean, D., and Dumont, M.: Snow depth mapping from stereo satellite imagery in mountainous terrain: evaluation using airborne laser-scanning data, The Cryosphere, 14, 2925–2940, https://doi.org/10.5194/tc-14-2925-2020, 2020.

McClung, D. M., & Schaerer, P. A. (1980, November). Snow avalanche size classification. In Proceedings of Avalanche Workshop (Vol. 3, No. 5).

Mergili, M., Fischer, J. T., Krenn, J., & Pudasaini, S. P. (2017). r. avaflow v1, an advanced open-source computational framework for the propagation and interaction of two-phase mass flows. Geoscientific Model Development, 10(2), 553-569.

Moner, I., Orgué, S., Gavaldà, J., & Bacardit, M. (2013). How big is big: results of the avalanche size classification survey. In Proceedings ISSW (Vol. 10).

Sampl, P., & Zwinger, T. (2004). Avalanche simulation with SAMOS. Annals of glaciology, 38, 393-398.

Shean, D. E., Alexandrov, O., Moratto, Z. M., Smith, B. E., Joughin, I. R., Porter, C., & Morin, P. (2016). An automated, open-source pipeline for mass production of digital elevation models (DEMs) from very-high-resolution commercial stereo satellite imagery. ISPRS Journal of Photogrammetry and Remote Sensing, 116, 101-117.

Sykes, J., Haegeli, P., and Bühler, Y.: Automated snow avalanche release area delineation in data sparse, remote, and forested regions, Nat. Hazards Earth Syst. Sci. Discuss. [preprint], https://doi.org/10.5194/nhess-2021-330, in review, 2021.

---

## Author Comment (AC2)

Anonymous Referee #2 (RC2)

Dear Anonymous Referee #2

Thank you for your constructive review of our paper.

We have provided answers to your questions and suggestions (your comments in italics) below:

> *The manuscript aims at providing insights into the DEM generation process by various approaches and the effect of DEM resolution on the subsequent numerical modeling. The manuscript is well written and well organized, and the methods are well explained. The work is interesting; however, some minor parts have to be improved.*
>
> 1. *Page 11, Line 285: "Holes were filled consistently by applying a max hole-filling threshold of 100 m2,.....". It is a little bit unclear to me. What is the smallest and largest size encountered? What does it mean by "max hole filling threshold of 100 m2", is this the largest hole size that can be filled? What technique is used? What are the advantages and drawbacks of the hole filling technique used? What impact did the hole-filling have on the modeling results? Also, any reference to available literature will be sufficient.*

Holes, or data voids, are primarily created when terrain is occluded from the sensor (in the case of TLS and SPM) or when there is low image contrast or clouds present in the image (in the case of SPM). Terrain occlusions are more acute with terrestrial sensors where the oblique angle to the terrain creates shadows behind topographic features and above-ground objects (Currier et al., 2019; Bühler et al., 2016). The high relief in our study site also created terrain occlusions to the satellite sensor, however this was in the terrain adjacent to the avalanche path and not in the path itself (see Figure 3). TLS-derived holes were mitigated by combing multiple scans taken from different locations in the study site into a composite point cloud before interpolation. Nonetheless occluded terrain remained in the TLS composite scan.

We took a conservative hole-filling approach to minimise the interpolation for areas lacking elevation measurements. We used a maximum 100 $m^2$ threshold for the occluded terrain that would be interpolated. This threshold came from a 5x5 cell window, based on the full-resolution SPM 2 m DSM. The aim was to fill holes in microtopography and avoid filling leaving larger holes where interpolation may affect the representation of the surface and modelling results. As in other topographic modelling applications where DEMs are conditioned to remove holes, pits and other interpolation artefacts and errors (Reuter et al., 2007), dynamic hazard models such as RAMMS require the use of a hole-filled DEM. Without hole-filling we would not have been able to conduct the sensitivity test with RAMMS. We believe the impact of hole-filling on the modelling results was minimal as we were targeting microtopographic features. However, we would advise caution for hole-filling large areas inside the RAMMS modelling domain as it creates a risk of over-smoothing the true terrain represented by the DSM. Coarser resolution DSMs will be less prone to the influence of hole-filling as they will already more smoothly represent the terrain compared with a high-resolution DSM.

Our hole-filling occurred during the interpolation from the source point cloud to the DSM with ASP's *point2dem* tool (Beyer et al., 2019; specific tool documentation: https://stereopipeline.readthedocs.io/en/latest/tools/point2dem.html). Holes are filled using neighboring cell values up the maximum number of cells specified.

As with Reviewer #1 comment, we will expand and clarify section 2.1.4 on hole-filling.

> 2. *Page 11, Line 272: In reference to comment #1, if there were holes encountered in the DEMs then how can be classified as "State of the art DEMs"?*

As discussed in response to previous comment, holes in high-resolution DSMs are common. We took a conservative approach to filling holes in microtopography while leaving the larger holes unfilled to avoid mis-representing the terrain since the study focused on best-representing the topography on which the avalanche flowed. However, we agree *state-of-the-art* is a subjective term and we will remove it from the paper.

> 3. *2: demarcate the release zone and show the runout direction.*

We will update Figure 2 to show the approximate delineation of the release zone and the runout direction.

> 4. *Page 14, Line 354: Please provide the respective DOD and corresponding information.*

We report on the results of the DoD in the Results section (Section 3.4, lines 478-486; Figure 8) which we feel is the appropriate place for the co-registration quality values, rather than in the Methods section. We will clarify the point in this section.

> 5. *4: it would be better to replace the figure with the time-lapse for the complete runout.*

We agree it would have been preferred to have a sequence of images showing the complete runout and final debris, however this is not available. The images in this figure came from a camera located down valley in safe zone where the final debris was not visible. We chose these frames as the core ejects over the lower cliff and splashes across the valley to coincide with the calibration RAMMS simulation.

> 6. *Please provide the RAMMS input parameters in a tabulated form.*

This is a good idea and was also requested by Reviewer #1. We will provide the RAMMS parameters in a table in the appendix.

**References**

Beyer, R., Alexandrov, O., and McMichael, S.: NeoGeographyToolkit/StereoPipeline: Ames Stereo Pipeline, v2.7, https://doi.org/DOI:10.5281/zenodo.3247734, 2019.

Bühler, Y., Adams, M. S., Bösch, R., and Stoffel, A.: Mapping snow depth in alpine terrain with unmanned aerial systems (UASs): potential and limitations, The Cryosphere, 10, 1075–1088, https://doi.org/10.5194/tc-10-1075-2016, 2016.

Currier, W. R., Pflug, J., Mazzotti, G., Jonas, T., Deems, J. S., Bormann, K. J., ... & Lundquist, J. D. (2019). Comparing aerial lidar observations with terrestrial lidar and snow-probe transects from NASA's 2017 SnowEx campaign. Water Resources Research, 55(7), 6285-6294.

Reuter, H. I., Nelson, A., & Jarvis, A. (2007). An evaluation of void-filling interpolation methods for SRTM data. International Journal of Geographical Information Science, 21(9), 983-1008.